# How Are Macro-Scale and Micro-Scale Built Environments Associated with Running Activity? The Application of Strava Data and Deep Learning in Inner London

**Hongchao Jiang** **, Lin Dong and Bing Qiu \***

School of Landscape Architecture, Nanjing Forestry University, Nanjing 210037, China
\* Correspondence: qiubing@njfu.edu.cn; Tel.: +86-138-5187-2804

**Abstract:** Running can promote public health. However, the association between running and the built environment, especially in terms of micro street-level factors, has rarely been studied. This study explored the influence of built environments at different scales on running in Inner London. The 5Ds framework (density, diversity, design, destination accessibility, and distance to transit) was used to classify the macro-scale features, and computer vision (CV) and deep learning (DL) were used to measure the micro-scale features. We extracted the accumulated GPS running data of 40,290 sample points from Strava. The spatial autoregressive combined (SAC) model revealed the spatial autocorrelation effect. The result showed that, for macro-scale features: (1) running occurs more frequently on trunk, primary, secondary, and tertiary roads, cycleways, and footways, but runners choose tracks, paths, pedestrian streets, and service streets relatively less; (2) safety, larger open space areas, and longer street lengths promote running; (3) streets with higher accessibility might attract runners (according to a spatial syntactic analysis); and (4) higher job density, POI entropy, canopy density, and high levels of PM 2.5 might impede running. For micro-scale features: (1) wider roads (especially sidewalks), more streetlights, trees, higher sky openness, and proximity to mountains and water facilitate running; and (2) more architectural interfaces, fences, and plants with low branching points might hinder running. The results revealed the linkages between built environments (on the macro- and micro-scale) and running in Inner London, which can provide practical suggestions for creating running-friendly cities.

**Keywords:** running activity; built environment; Strava; street view images (SVIs); deep learning; semantic segmentation; spatial inequality; Inner London

## 1. Introduction

Promoting physical activity may significantly improve public health and well-being and further contributes to the sustainability of cities and society [1–4]. Running has multiple mental and physical benefits [5,6]. Since 1960, running has gradually gained worldwide popularity [7]. While numerous studies have addressed which urban built environment features contribute to walking and cycling [2,4,8,9], few have investigated how the built environment affects running activities [5,10]. Since running is different from those physical activities (e.g., in terms of speed, spatial scope, and sensory experience) [11], we cannot freely assume that an environment that is attractive for walking, biking, or team sports will be equally attractive for runners [10]. Therefore, it is necessary to better understand the relationships between the built environment and running behavior.

There are three main gaps among the prior running studies. First, previous running environment studies have had limitations on data sources [12]. The data collection methods of traditional studies were mostly questionnaires [5,13], participant observation [14], and diary interviews [15], which are time-consuming and laborious. In addition, the sample size was rather limited, so the large-scale analyses of the running's spatial rules were

inadequate. In recent years, the emergence of GPS crowdsourcing tracking data provides a new data source for running activities, which can record the behavior patterns of runners in a large range and provide a new opportunity for the research of running patterns at the urban scale.

Second, only a few studies have focused on the impact of limited built environment factors (variables) on running [5,11,13] and lack comprehensive and systematic measurement of the built environment [6], so some macro-scale features that may potentially affect running may be overlooked. In the field of urban research, the 5Ds framework proposed by Cervero et al. [9,16] has been widely recognized by researchers and professionals to classify and measure the attributes of the built environment. It has been widely applied in walking and cycling studies as a representation of the five aspects of the macro-scale built environment [2,4,17,18]. The application of the 5Ds framework may be an effective way to classify and measure the macro-scale built environment that may affect running.

Third, the micro-scale built environment features based on eye-level views have been shown to have a significant impact on human physical activities [6,11]. However, due to the limitations of technology, the influence of street-level elements on running activities is rarely discussed in running environment studies. New data sources and technologies have enabled the study of the influence of the built environment on running more deeply and broadly. The combination of SVI data and DL technology provides new opportunities to understand the relationship between the built environment and running at the micro-scale by quantifying the street environment at the approximate eye level of runners [15,19].

Against this backdrop, this study explored the spatial patterns of city-wide running activities and scrutinized the influence of building environment features at different scales on running in Inner London using multi-source data, SVIs, and DL. We will mainly explore the following four questions:

(1) How and which macro-scale built environment attributes based on the 5Ds framework influence the running amount in Inner London?
(2) How and which micro-scale streetscape features based on CV influence the running amount?
(3) How do micro-scale streetscape features complement or conflict with macro-scale built environment indexes?
(4) Does the running amount in Inner London (as geospatial data) have spatial dependence effects?

To test these questions, we (1) used secondary data from the Strava Heatmap (SH) to obtain the distribution of running (two-year accumulated GPS running tracks) in Inner London; (2) extracted multi-source data to measure the macro-scale built environment features based on the 5Ds framework; and (3) acquired a large number of Google Street View images (GSVs) using DL technology (a semantic segmentation framework—Pyramid Scene Parsing Network, PSPNet) and CV to measure the microscopic built environment features at a level similar to a runner's line of sight. We combined the two different scales to measure the linkages between running and built environments comprehensively. Furthermore, we explored the potential association between macro- and micro-scale environment features. Finally, based on a traditional regression analysis, we established a spatial regression model to measure the effect of spatial dependence.

The contributions of this study can be summarized as follows. First, we attempted to use a semi-open data source, SH, to investigate runners' route preferences and spatial clustering effects at the city scale in Inner London. Second, our study adds to the running literature by combining the traditional 5Ds framework with the microscopic built environment features from runners' sight, which allows for a comprehensive analysis of the influence of built environments on running. Third, we explored the internal correlation between macro-scale built environment features and micro street-level characteristics and compared their degrees of contribution to the explanatory power of running activities, which has not been evaluated in running studies. Finally, the effect of spatial autocorrela-

tion has rarely been considered in previous studies; therefore, we used the SAC model to take spatial dependence into account.

## 2. Review of the Literature

Running can be seen as an interaction between the body, senses, and environment [20]. There is strong evidence that various built environment features can influence running. For example, Bodin and Hartig [21] suggested that outdoor environments promote running by regulating the runner's psychology. The authors of [22] found that the physical environment had an important impact on runners' running performance. Academics and policymakers are becoming increasingly aware that well-designed public spaces can help create pleasant urban environments that are attractive to runners. Both the macro- and micro-scale built environment features had been discussed in previous running studies, but rarely systematically classified.

### 2.1. Macro-Scale Built Environment and Running

In the field of urban research, a popular framework for assessing the macro-scale built environment is the 5Ds (density, diversity, design, destination accessibility, and distance to transit). Three dimensions called the 3Ds (density, diversity, and design) were first put forward to express the urban environment by Cervero and Kockelman [23]. Two more dimensions (destination accessibility and distance to transit) were later proposed as a complement to form the 5Ds [9]. Ewing and Cervero [16] found it useful to use the D variables to organize the empirical literature and provide order-of-magnitude insights. A few scholars have recently attempted to apply the 3Ds or 5Ds framework to the running field. Yang et al. [12] took the lead in using 5Ds to measure the impact of the built environment on the strength index of running and cycling. On this basis, we used the classification dimension of the 5Ds classical urban theory framework to separately discuss the macro-scale built environment factors involved in previous running studies.

Density is recognized as one of the most essential and frequently used built environment variables [24]. The common measurements of density are the population density and building density [25]. Many studies have shown a significant correlation between population density and physical activity, but researchers differ as to how the population density affects (promotes or hinders) running. Ettema [11] indicated that a densely populated urban area decreases the enjoyment experienced from running due to the hindrance of public traffic. Huang et al. [6] found that running satisfaction was not related to the population density. In terms of building density, Yang et al. [12] revealed that the residential building density and floor area ratio have either negative or insignificant effects on running indices in Chengdu. They explained that Chengdu is a high-density city that is experiencing constant urban expansion, so the impact of the residential building density in Chengdu on physical activity may differ from that observed in Western countries.

In the dimension of diversity, Troped et al. [26] argued that a higher level of the land-use mix can promote moderate–vigorous physical activity, but Yang et al. [12] concluded that the land-use mix has insignificant effects on the running activity of residents, which largely contrasts with the evidence gathered from previous studies.

The design dimension has been most discussed in previous running studies. Many studies have shown that runners prefer to be in nature and away from the hustle and bustle of the urban environment. The authors of [21,27] found that runners prefer green environments more than urban settings due to their attractiveness and lack of association with daily troubles. The authors of [28,29] indicated that runners tended to run inside parks and to stay away from traffic and intersections to avoid a fragmented experience. In addition, a safer running atmosphere was also proven to be the reason for the popularity of various running paths in structured interviews (*N* = 546) that were conducted by Borgers et al. [30]. Moreover, street design characteristics, such as the traffic conditions, tree density, street light density, terrain slope, top-down greenness, blue space density, length of the street

segment, and the twists and turns of the path, have all been shown to affect the running satisfaction or consistency of the running rhythm [6,31,32].

Destination accessibility was recognized as one of the most fundamental factors for urban physical activity. This dimension was less involved in the study of running, but in the field of walking, the spatial design network analysis (sDNA), a space syntax tool, was often used to measure spatial connectivity [25]. The analysis of betweenness indicates the "through-movement" possibility, which means the potential of each street unit to be chosen for physical activity. Therefore, it can be used to predict the most easily accessed streets [33]. Furthermore, angular distance-based accessibility values have been proven to be well correlated with observed human–vehicle behavior distributions [34].

Distance to transit is usually measured as the shortest network distance from an origin to a nearby transit stop [16]. Public transit use is thought to encourage more physical activities. Huang et al. [6] found that running routes were more satisfying when they had more public transport nodes.

In conclusion, a range of running studies have investigated the relationship between macro-scale built environment variables and running, but few have discussed this 5Ds framework systematically. Therefore, there is a need to measure macro-scale built environment features comprehensively and systematically and their relationships with running.

### 2.2. Micro-Scale Built Environment Features and Running

Micro-scale built environment features mainly refer to the elements and features that pedestrians can directly perceive on the streets [35,36]. It can be seen from the previous literature that some street-level microscopic built environment factors do have an impact on running. Huang et al. [6] found that the eye-level greenness of streetscape features had a positive impact on running satisfaction. The authors of [37] noted that different surfaces affect the effort required (e.g., smooth pavement or grass vs. uneven pavements, muddy paths, and holes) and may increase the probability of injury. Other street environmental features, such as vehicles, pedestrians, cyclists, and street animals (such as unreleased dogs), have also been found to be associated with the running experience [11,38].

However, due to the limitations of technology and measurement methods, previous studies tend to involve only a few street-level elements. Few researchers have systematically considered the association between the microscopic built environment elements and running activity, so the influence of other street elements on running activity is unknown.

Fortunately, SVIs, as a large-scale data source, have been used to examine visual features from a near-human perspective that top-down data sources (e.g., aviation, satellite data) cannot provide [6,8,39]. In addition, DL technology has become a vital tool in the semantic segmentation of SVIs and has made continuous improvements to the accuracy of prediction. Thus, this provides a new opportunity to examine the relationship between the microscopic built environment in finer detail and the running behavior of a larger area. In the research of walking or cycling, some scholars have used the emerging image segmentation technology to measure the physical features of the built environment at the street level and explored the impact of these microscopic street features on physical activity, which has achieved good results [2,3,40]. However, this new measure has rarely been used in studies of running activity. Dong et al. [32] recently investigated the impacts of the physical streetscape and perceptions on running using Resnet (a deep residual network) and CV. They found that the streetscape quality is an important running-influencing factor for running in Boston. The results indicated that running has positive associations with the pixel ratios of vegetation, sky, terrain, sidewalk, wall, fence, person, etc., while it is negatively related to the pixel ratios of motorcycle and traffic light. Despite this, studies that simultaneously assess the impact of macro-scale variables and micro-scale street elements on running under a large geographic area remain little known.

### 2.3. The Rise of Crowdsourced Data for Running

In recent years, there has been a boom in apps that use GPS devices (e.g., mobile phones, fitness watches) to track users' behavior and record personal fitness processes. These crowdsourced data can increase the number of observations and show route preferences in human movements via massive movement tracking [41]. In terms of running, Strava is the largest platform, which includes Strava Metro and SH [42]. Strava Metro provides detailed data on running users, but only for certain partners. However, SH is freely available to registered users and contains trillions of data points at the street level [43,44]. SH can accurately show the track points and routes of an athlete's actual run, as well as the cumulative number of runs on each street using color saturation, which represents the GPS point density of the users. The higher the heat value, the greater the cumulative number of runs conducted at that location [42].

Researchers have realized the value of SH in academic studies to measure hot spots and routes at a large scale and to extract spatial patterns of movement. For example, Rice et al. [45] extracted GPS track data from SH by overlaying street networks in GIS. The method offers a promising approach to obtaining data on physical activity. In addition, Havinga et al. [46] extracted the average "heat" intensity (the 18 m circular areas around the midpoint of each road) from SH to represent running activity data.

Recently, Cahill and Woods [47] used the same method as Rice et al. (2019) [45] to take computer screenshots of SH running and cycling maps using the Microsoft Snipping Tool. They obtained a total of 24 screenshots and used GIS for the reclassification of pixel values (as the use intensity of runners and cyclists on forest roads and hiking trails). They proposed an innovative way to explore recreational use preferences, despite its heavy reliance on pixelated data from SH. Dong et al. [32] also used SH screenshots to obtain the value of the running amount on each street segment in Boston to explore route choices. Yang et al. [12] constructed spatial regression models to analyze the relationship between the built environment and running and cycling using SH in Chengdu, China. Since Strava crowdsourced data have many advantages in terms of data collection, SH was used in this study to measure the running amount.

## 3. Data and Methods

### 3.1. Study Area and Analytical Framework

London is the capital, and largest, city of the United Kingdom. The general topography of London is relatively flat with many outdoor running trails that are suitable for regular jogs and long runs. Furthermore, the latest mayor's strategy—"Sport for all of us"—has encouraged greater participation in sports and physical activities to improve health and well-being and build links between diverse communities. All these give London a unique running atmosphere. Our study area was Inner London (Figure 1). Inner London is the interior part of Greater London and is surrounded by Outer London. It covers 123 square miles and has a population of 3,536,000. It is a densely populated area of London. The boundary of Inner London was downloaded from the Greater London Authority (GLA).

The framework is composed of four stages (Figure 2). First, we collect multi-source datasets on a cloud server. Second, we extract the running raster value as the dependent variable and a series of features that may influence running as independent variables in 20 m, 50 m, and 100 m buffer zones. These independent variables are divided into three groups: macro-scale built environment characteristics (5Ds), micro-scale built environment characteristics, and control variables. Then, we use the extracted features for correlation analysis and use the ordinary least squares (OLS) method to explore the impacts of multiple variables on running and the relationship between each independent variable. Finally, we use the SAC model to test the spatial dependence and solve the bias caused by the spatial autocorrelation effect.

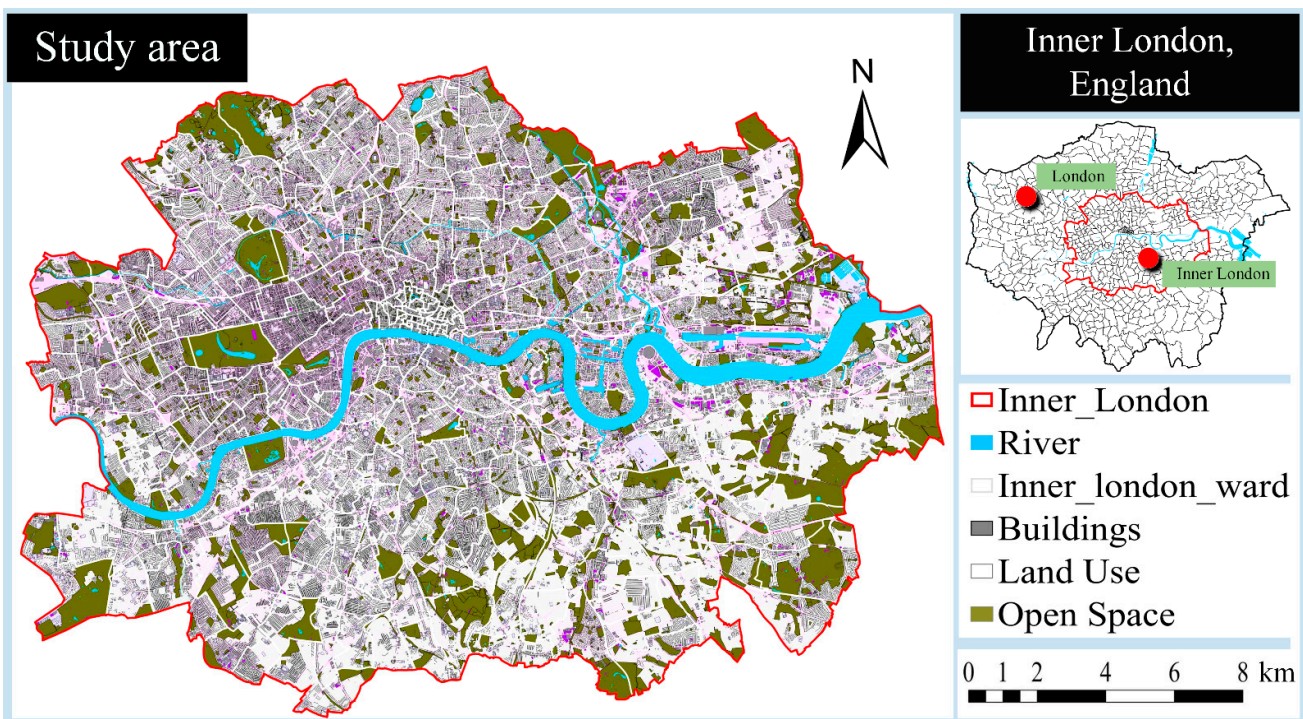

**Figure 1.** The extent of the study area, Inner London, England.

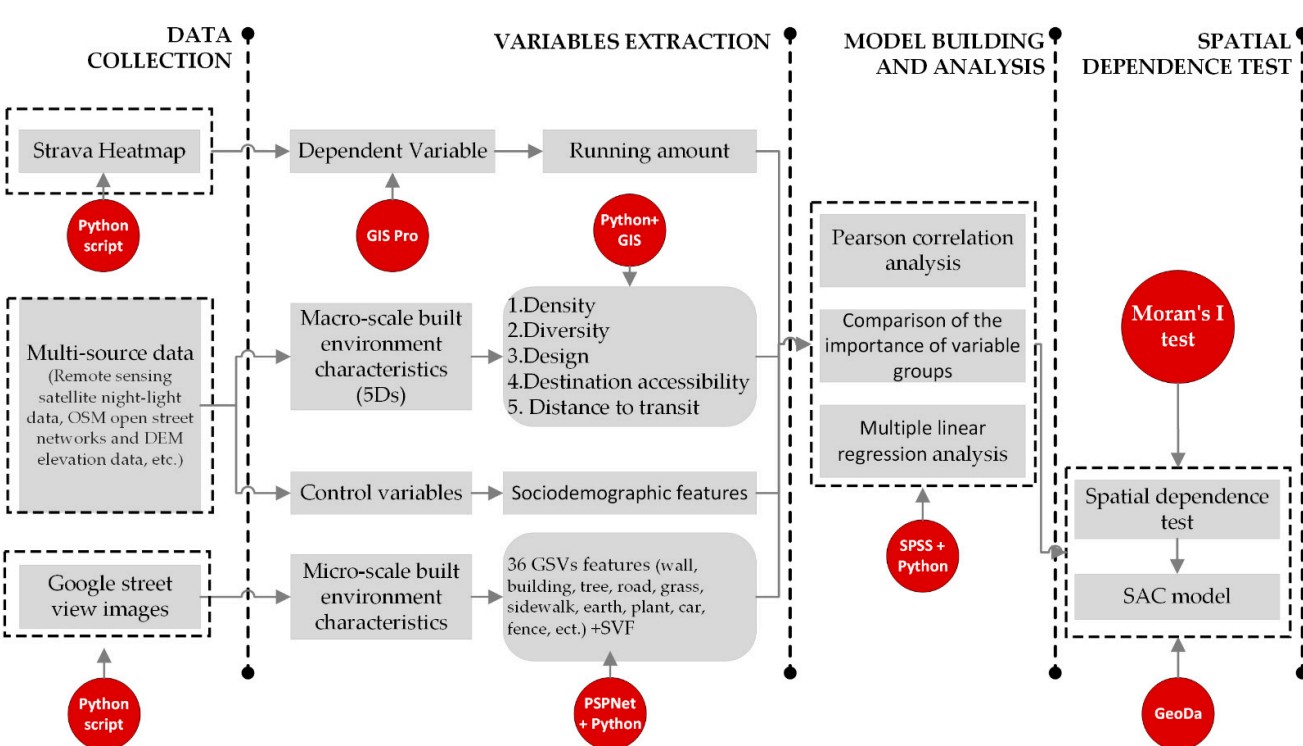

**Figure 2.** The analytical framework of this research.

*3.2. Data Collection and Variables Extraction*

3.2.1. Dependent Variable: Running Amount

According to Strava's official documentation, SH shows public activity data aggregated over the past two years, reflecting accumulated public activity [44,48]. In Greater London, there are 4559 run-themed clubs in total. Among these, there are 30 running clubs with more than 10,000 members and 79 running clubs with 1000 to 10,000 members. The Weekly

5kClub Run alone has 72,133 running members. The cumulative results of the running trajectories of these runners over two years could reflect the local running hotspots and the spatial preference patterns of runners to a certain extent. We wrote a Python script to climb the high-definition thermal raster map of running from SH, in contrast to previous studies [12,45,47] that captured image raster data using computer screens. We obtained 594 tiles (512 × 512 pixels in each picture) and assembled a complete raster image of Inner London (Figure 3). All the tiles of the raster data that we captured were at the same map zoom level; therefore, our method is more accurate and reliable than previous screenshot methods. Since the visual map of the Strava platform is based on the OpenStreetMap (OSM) street networks [47], we downloaded OSM street data and superimposed it to extract running data using the tool "Zonal Statistics as Table" in ArcGIS Pro software (version 2.8.6, Esri, Redlands, CA, USA). Then, we extracted and calculated the raster value of each street to obtain the mean value.

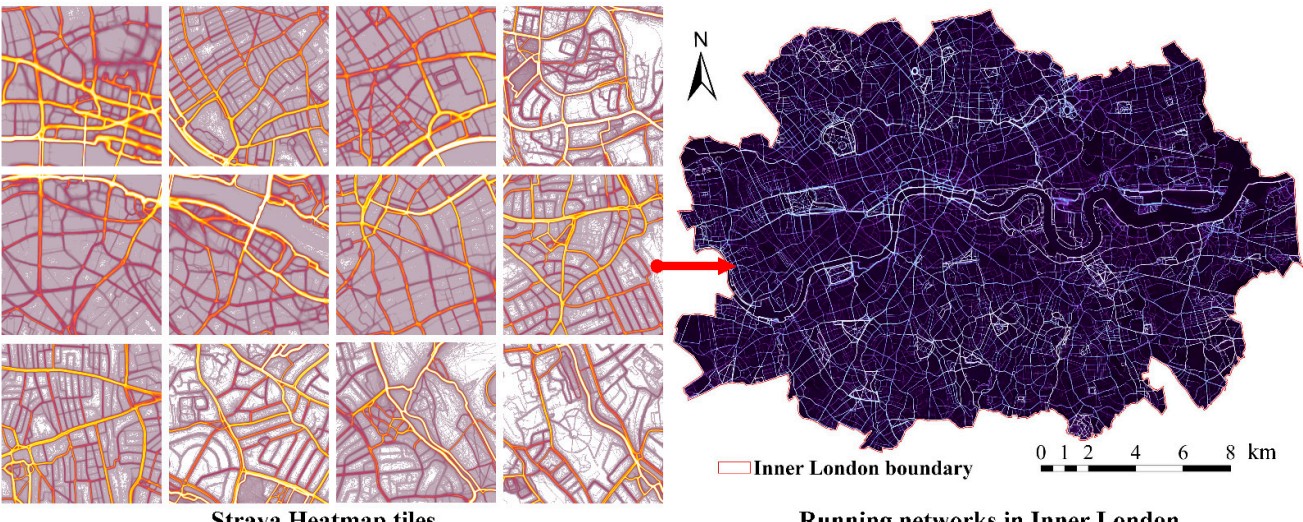

**Strava Heatmap tiles**  **Running networks in Inner London**

**Figure 3.** Tiles of 512 × 512 pixels captured from SH and the complete running heat map of Inner London made up of 594 tiles.

To more comprehensively cover the street environment of Inner London and to obtain the geographical coordinates of GSVs, sample points were generated along each street segment at 100 m intervals [49,50]. Finally, 48,286 sample points were obtained. The mean raster value was regarded as the proxy of the cumulative running number (running amount) of each sample point, and as the dependent variable for this study.

### 3.2.2. Macro-Scale Built Environment: 5Ds

Expanded from Cervero and Ewing, this study adjusted the 5Ds variables based on the characteristics and data availability of the built environment in Inner London. Most of the data were collected from the Office for National Statistics (ONS), GLA, and Transport for London (TFL) between 2018 and 2021. Table 1 shows the sources, data description, and processing method of all variables.

It is important to emphasize that this study used angular betweenness (BtA) as a measure of road network accessibility. The radius is essential in sDNA analysis. The results of the accessibility of the analysis of different radii correspond to road selectivity for travel behavior at corresponding radii distances [25,51]. According to Strava's annual summary reports for 2021 and 2020 [52,53], the average running distance of Strava users worldwide in 2021 was 6.28 km, and the average running duration was 0:38:48 min. In 2020, the average global Strava user ran 6.3 km in 0:38:48 min. The average distance run is 800 m per five minutes. Therefore, the two distances of 800 m and 6.3 km were selected as the analysis radii of running in our study.

**Table 1.** The data sources and extraction methods of all independent variables.

| Variables | Data Source | Data Description and Extraction | Reference |
|---|---|---|---|
| Control variables | | | |
| Age groups | | | |
| Pop0to17, Pop18to44, Pop45to59, Pop60to74, Pop over75 | ONS | Add up the population for each age from 0 to 90+ at the London ward level | Sarkar et al. [33] |
| Per capita income 2019 | ONS | The amount of money of each person at the London ward level | Andersson et al. [54] |
| Macro-scale built environments 5Ds | | | |
| Density | | | |
| Population density 2020 | ONS | Extracted at London ward level | Cervero and Kockelman [23] Li et al. [17] |
| Job density 2019 | ONS | Extracted at London borough level | Andersson et al. [54] |
| Building density | OSM | Building area divided by 20, 50, and 100 m buffer area | Cervero et al. [9] Rebecchi et al. [4] |
| Diversity | | | |
| Street type | | | |
| Trunk road, primary road, secondary road, tertiary road, residential street, living street, pedestrian street, cycleway, footway, service street, track, path | OSM | Each street corresponds to a street type (0. no; 1. yes). The street networks downloaded from OSM include a total of 20 road types. According to the definition of OSM road classification [55], eight road types, such as bridleway, steps, unclassified, etc., were removed [49,56] | Ito et al. [3] Sultan et al. [56] Li et al. [49] |
| POI entropy | OSM | Calculate Shannon–Wiener index | Yang et al. [12] |
| Design | | | |
| Design: street amenities | | | |
| Open space area | Planning Constraints Map | Open space area divided by 20, 50, and 100 m buffer area | Cervero et al. [9] Yang et al. [12] |
| Canopy density | Breadboard Labs and GLA | A high-resolution map (25 cm per pixel) of the tree canopy cover of London was produced from aerial imagery using CV and DL (accuracy 94.87%). We calculated the canopy density and extracted it to each sample point | Cervero et al. [9] Sarkar et al. [33] |
| Number of intersections | OSM | 20, 50, and 100 m buffer zones | Lee et al. [57] Rebecchi et al. [4] |
| Number of traffic lights | OSM | 20, 50, and 100 m buffer zones | Cervero et al. [9] |
| Number of parking lots | OSM | 20, 50, and 100 m buffer zones | Rebecchi et al. [4] Cervero et al. [9] |
| Maximum speed | OSM | Extract the maximum vehicle speed of each street to each sample point | |
| Street length | OSM | Calculate the length of each street using GIS | Wang et al. [58] |

**Table 1.** *Cont.*

| Variables | Data Source | Data Description and Extraction | Reference |
|---|---|---|---|
| **Design: safety** | | | |
| Number of crimes | Metropolitan Police Service | Calculated the total number of crimes (the last 24 months) in the ward-level area | Cervero et al. [9] |
| Number of traffic accidents | Department for Transport | The number of traffic accidents (2020) within the 20, 50, and 100 m buffer zones of each sample point were calculated | Cervero et al. [9] |
| Number of fires | London Fire Brigade | Calculated the number of fires (the last 5 years) in each ward-level area | Schuurman et al. [5] |
| **Design: level of street pollution** | | | |
| Annual mean NO2 | TFL | The dataset (2019) includes ground-level concentrations of annual mean NO2 in $\mu g/m^3$ at a 20 m grid resolution. We extracted the annual mean concentration of each sample point | Larkin et al. [59] |
| Annual mean PM2.5 | TFL | Same as above | Same as above |
| Street noise pollution level | Department for Environment, Food and Rural Affairs | The noise pollution dataset shows the annual mean noise level from roads and rails We superimposed and extracted the values to each sample point | Schuurman et al. [5] |
| **Design: street environment attributes** | | | |
| Street slope | National Aeronautics and Space Administration DEM data | Used the "slope" tool in GIS and extracted values to each sample point (12.5m accuracy). | Ito et al. [3] Cervero and Kockelman [23] |
| Night-light intensity | LJ1-01 remote sensing satellite data | Superimposed the imaging data (from 19 June to 3 October 2018, 130 m resolution), calculated the mean input radiance value of night light, and extracted it to each sample point | Yang et al. [12] |
| Annual mean temperature | GLA, Landsat-8 Thermal Satellite data | Recorded London's major daytime summer hotspots (30 m resolutions). We extracted the temperature value to each sample point | Balaban and Tunçer [60] |
| **Destination accessibility** | | | |
| BtA800, BtA6300 | Space syntax tool | Calculated and extracted angular-distance-based accessibility values | Sarkar at al. [33] Tang at al. [25] |
| **Distance to transit** | | | |
| The public transport accessibility levels (PTALs) | TFL | A precise measurement of the density of the public transport network at any location within London (100 m accuracy). We extracted the density of the public transport network to each point | Cervero et al. [9] Ewing and Cervero [16] |
| **Micro-scale built environments** | | | |
| Wall, building, tree, road, grass, sidewalk, earth, plant, car, fence, signboard, awning, streetlight, van, ashcan, railing, person, minibike, chair, sculpture, bicycle, column, bridge, water, fountain, windowpane, mountain, ceiling, booth, sofa, lamp, skyscraper, lake, bulletin board, desk, pier | 40,290 GSV images | PSPNet semantic segmentation framework using Python script | Qiu et al. [61] Dong et al. [32] |
| Sky view factor (SVF) | 40,290 GSV images | Generate fisheye and calculate SVF values using Python script | Li et al. [62] Xia et al. [63] |

### 3.2.3. Micro-Scale Built Environment

We obtained pixel ratios of 37 physical features of streetscapes using SVIs and CV in three stages (Table 1).

(1)  Collection of GSVs. To obtain GSVs of the 48,286 sample points, based on the longitude and latitude of each point, a Python script was used to download the latest GSV panoramas from May to October [64] using the Google Street View Static API. Finally, we collected 360° panorama images of 40,290 sample points along streets with a size of 2048 × 1024 pixels.

(2)  Semantic segmentation. The pixels of physical features of the panoramas were extracted using PSPNet and model training based on the ADE20K dataset, which is tailored for semantic urban scene understanding [65]. Moreover, PSPNet could accurately parse the scenes with complicated elements and has been used by many relevant urban studies [61,66]. The prediction accuracy of PSPNet in this study was 93.4%.

However, due to the image distortion of panorama images in the upper portions and lower portions, it is not suitable to extract the elements directly [40,67,68]. Tsai and Chang [67] mentioned that the center of the panoramic images was less distorted. They proposed to locate visual elements on the central portions of ±30° in pitch according to the vertical field of view of the camera lens. We adopted this method in our study, as is shown in Figure 4. The parts at approximately human eye level with a lower degree of distortion were used in the semantic segmentation process.

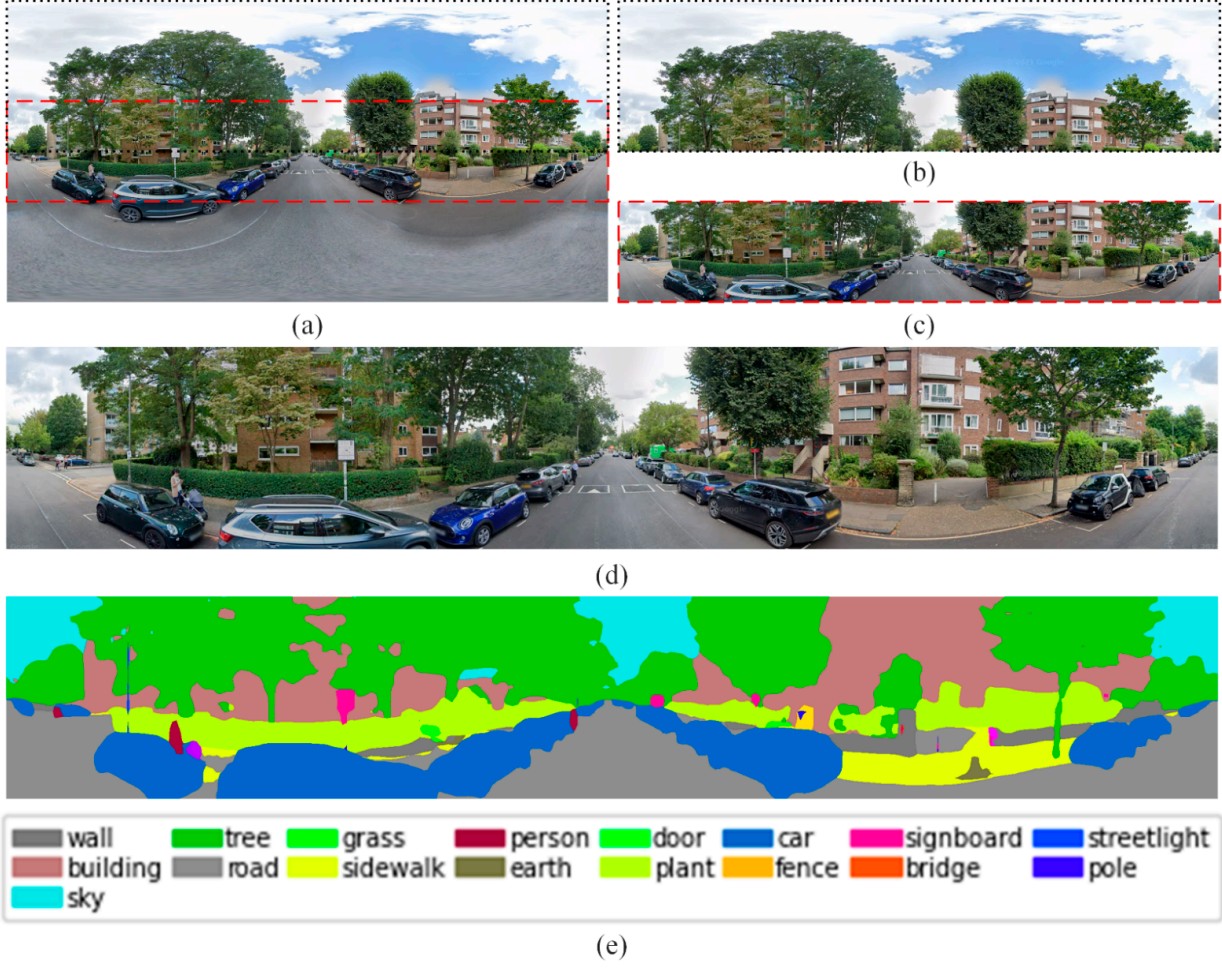

**Figure 4.** Semantic segmentation based on PSPNet. (**a**,**c**,**d**) The part within the red dotted line is the part where we extracted the street-level built environment characteristics; (**a**,**b**) the part within the black dotted line is where we used to calculate SVF values; (**e**) the result of semantic segmentation.

(3) Calculate the sky view factor. SVF is an important indicator to evaluate the openness of streets, which reflects the architectural form and the thermal comfort of the micro-scale street building environment along the street [63]. It can quantify the level of enclosure of street canyons and can be calculated as the ratio of the visible sky area to the total sky area at one point on a street (from 0 to 1), where 1 indicates an entirely open area and 0 means a completely covered space [62]. The method developed by Xia et al. [63] can generate fisheye (hemispherical) images to quantify street-level SVF values based on the DL model. In this study, we wrote a Python script to estimate the sky area using the upper part of the panoramic images [65] (Figure 5). Next, SVF values were calculated using Equation (1), which has also been applied and proven to be effective by Cao et al. [69].

$$SVF = \frac{Area_{s\_i}}{Area_{t\_i}} \times \frac{4}{\pi} \tag{1}$$

where, $Area_{s\_i}$ refers to the sky area pixels in the image taken in the *i*th sample point and $Area_{t\_i}$ refers to the total pixels of the image taken at the *i*th sample point.

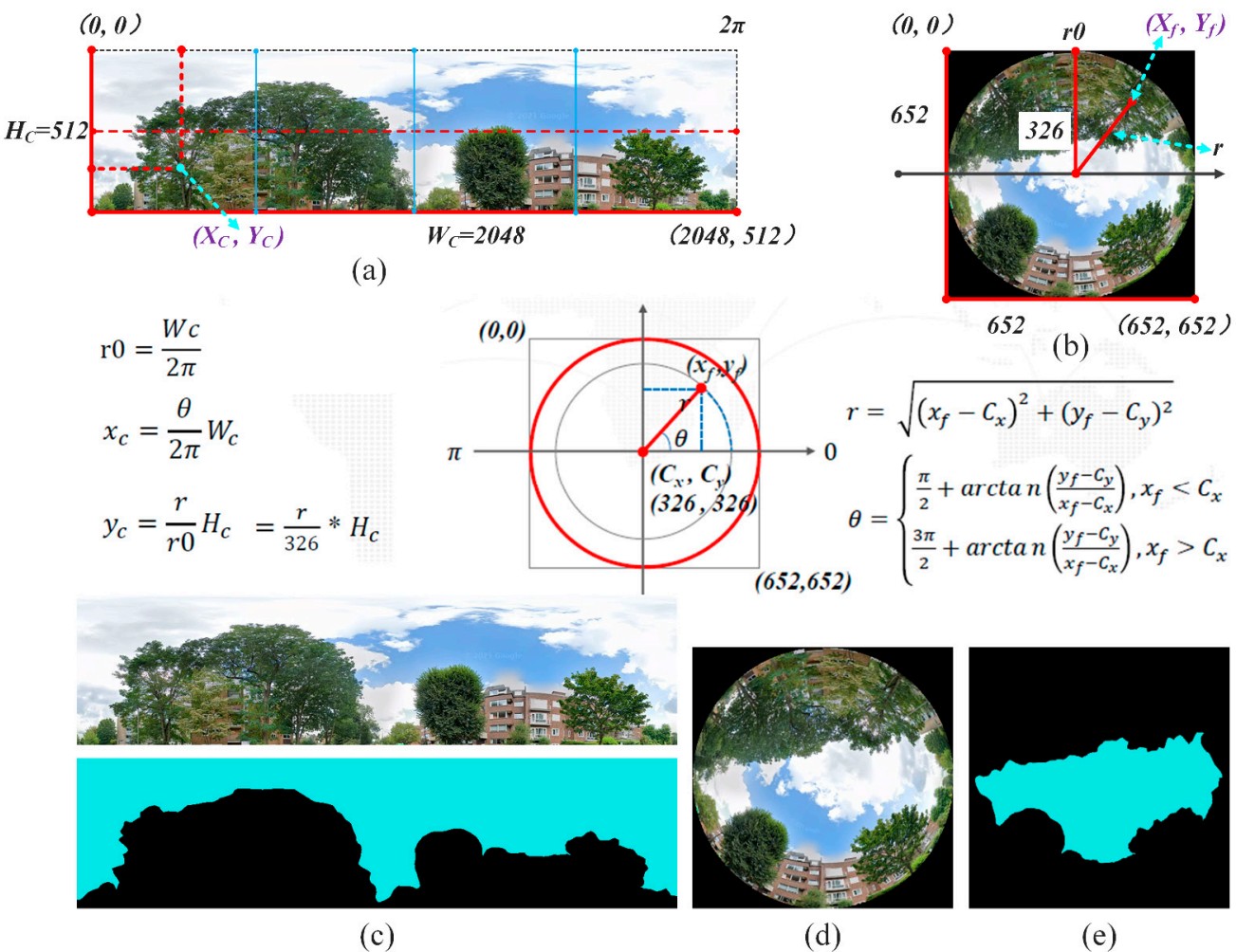

**Figure 5.** Calculate SVF. (**a,b**) The calculation process to generate a fisheye image; (**c**) the original image and the semantic segmented image of part of the sky; (**d**) the fisheye image generated using a Python script; (**e**) the pixels of the sky in the fisheye image that were extracted.

### 3.2.4. Control Variables

The control variables include age groups and per capita income. The age groups dataset (2020) provided the ward-level population for every age from 0 to 90+. We divided and calculated the population into five groups (0–17, 18–44, 45–59, 60–74, and over 75).

### 3.3. Correlation Analysis and OLS Analysis

We first conducted a Pearson correlation analysis on all variables and dependent variables. The variables unrelated to the running amount were eliminated ($p > 0.05$). Then, we compared the relative importance of three variable groups (control variables, macro-scale- and micro-scale built environment features) on the running amount using OLS.

Considering the modifiable areal unit problem, which is a statistical bias that leads to the impacts of aggregation of geographic data on the analysis results [70], the OLS models were established at a range of 20 m, 50 m, and 100 m buffers [6,10,68], respectively, to verify the stability of the models. Then, the buffer with the better result would be used for the spatial model analysis.

To compare the contribution of the three variable groups to the explanatory power in the model, especially the relationship between micro variables and macro variables, four OLS models were generated as follows:

OLS Model1 Running amount (Y)~Control variables
OLS Model2 Running amount (Y)~Control variables + Macro-scale 5Ds
OLS Model3 Running amount (Y)~Control variables + Micro-scale
OLS Model4 Running amount (Y)~Control variables + Macro-scale 5Ds + Micro-scale

### 3.4. Spatial Dependence Test and the Spatial Model

Due to the existence of spatial autocorrelation, there will be deviations in OLS model fitting [18,61,64,66]. The spatial autocorrelation effect can be tested by Moran's I [64]. If the Moran's I shows a significant correlation, it indicates that the OLS model has spatial autocorrelation. Therefore, we needed to establish a corresponding spatial model for further analysis to ensure the accuracy of the regression results according to relevant indicators, such as robust Lagrange multiplier (lag) and robust Lagrange multiplier (error), in OLS [50,61]. Moran's I tests and the construction of the spatial model were carried out in GeoDa 1.20.

In our case, the robust Lagrange multiplier indicates the existence of both spatial lag and error effects. Therefore, a SAC model with both a spatially lagged dependent variable (*W*y) and a spatially autocorrelated error term (*W*μ) [61] was used. It was calculated as follows:

$$y_i = \rho \sum_{j=1}^{n} W_{ij} y_j + \beta x_i + \mu\mu = \lambda W\mu + \varepsilon_i \qquad (2)$$

where, $y_i$ is the level of running amount in point $i$; $y_j$ is the level of running amount in point $j$; $\rho$ is the spatial autocorrelation coefficient; $w_{ij}$ is the spatial weight matrix; $\beta$ is the coefficient of the variables we chose; $x_i$ is the value of the variables in point $i$; $\mu$ is a vector of the spatial autoregressive error term; $\lambda$ is the coefficient of the spatial dependence in error terms; $\varepsilon_i$ is the error term; and n is the number of sample points in Inner London.

## 4. Results

### 4.1. Descriptive Statistics

Figure 6a shows the running amount in the OSM street networks extracted from the SH using GIS. Figure 6b illustrates the spatial distribution of the raster value of 40,290 sample points in Inner London. The value of the running amount varies from 0 to 255, with a mean value of 78.736 and a standard deviation of 65.165. From the perspective of the spatial distribution of running hot spots, the areas with frequent running in Inner London are relatively uniform at the regional level. One obvious phenomenon is the concentration

of running tracks along the banks of the River Thames, as well as near large parks or open spaces (e.g., Hyde Park, Regent's Park, Battersea Park, Victoria Park, Greenwich Park, etc.). In addition, some main streets also showed high raster values. Specifically, as is shown in Figure 6c, positions B and G are high-frequency running areas, as they are close to large open spaces and surrounded by greenery. F is located on the River Thames, and the area around the river has high "heat". C, D, E, and H are all located in residential areas that show moderate to low levels of running. In addition, A was located near the highway, which showed a lower running frequency. To further explore the distribution rules and influencing factors of running, we conducted the following series of analyses. The descriptive statistics for all variables are shown in Table A1.

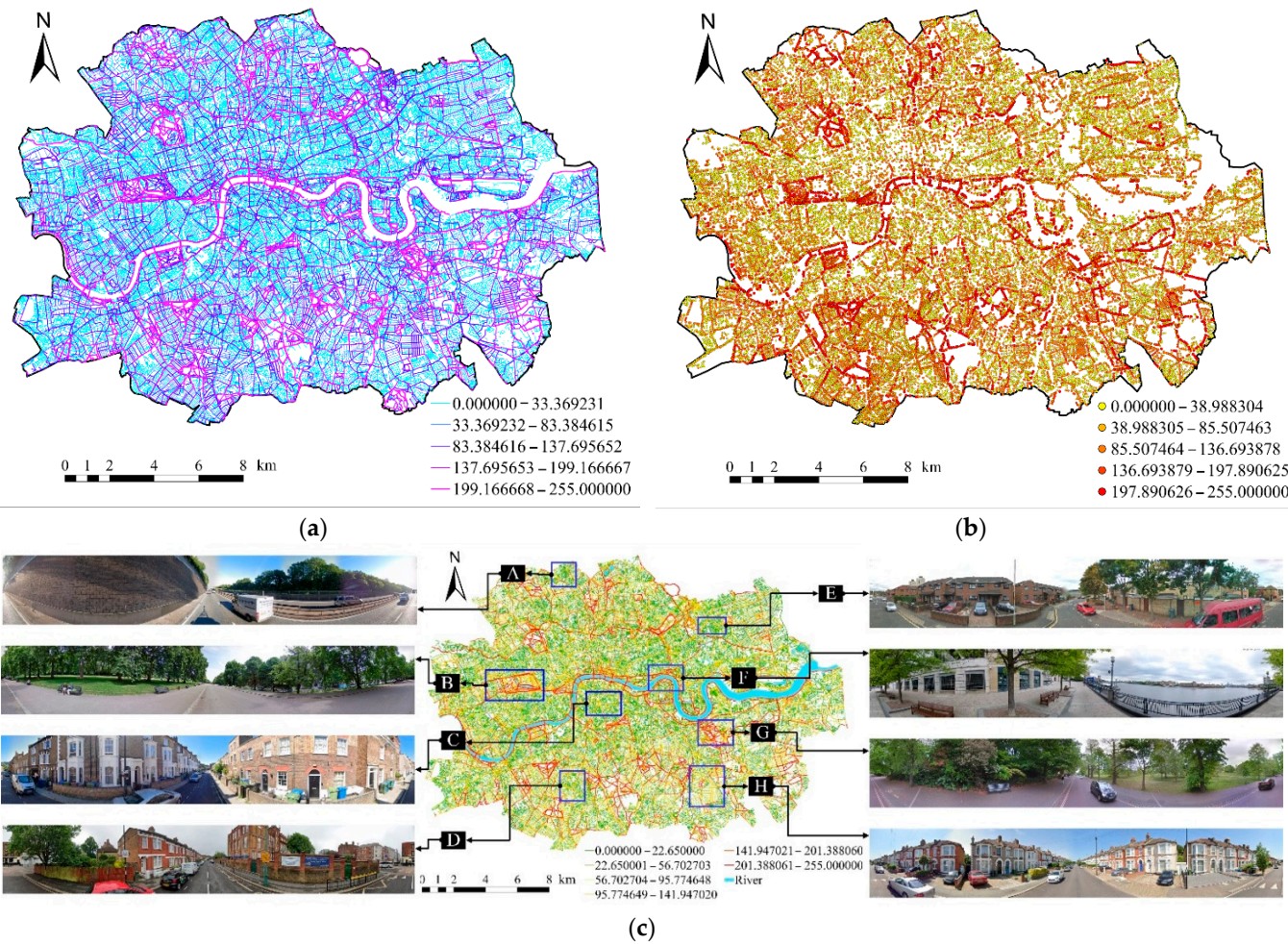

**Figure 6.** The value of the running amount and the spatial distribution of 40,290 sample points. (**a**) The raster value in the OSM street networks extracted from the SH in Inner London; (**b**) the spatial distribution of the running amount of 40,290 sample points; (**c**) the street scenes of some representative areas, the A–H area corresponds to a street scene, respectively.

### 4.2. Correlation Analysis

After the Pearson correlation analysis for all independent and dependent variables, in the 20 m buffer zone, 13 variables showed no significant correlation ($p > 0.05$), namely pop 60 to 74, living street, number of parking lots, night-light intensity, and nine variables of semantic segmentation (van, chair, bicycle, column, ceiling, sofa, lamp, bulletin board, and desk). In the 50 m and 100 m buffer zones, the insignificant variables were the same as the above variables except that the number of parking lots became significantly correlated with the running amount. The irrelevant variables were removed in the subsequent analysis.

### 4.3. OLS Results and the Relative Importance of Variable Groups

We standardized the independent variables using the Z-score standardization method. By using this approach, we standardized variables with different observation scales into the same scale to eliminate the dimensional differences caused by different units. The mean of the newly generated variables is 0 and the standard deviation is 1, but they keep the distribution trend of the original data.

Table 2 shows the relative importance of the macro-scale, micro-scale, and control variable groups. In the 20 m, 50 m, and 100 m buffer zones, the ranking results of the importance of the three variable groups were consistent, and the F statistics were all significant. Furthermore, to avoid the multicollinearity problem of OLS models, we took a variance inflation factor (VIF) of less than 10 as the standard [61,66]. The variables with VIF > 10 were removed.

**Table 2.** The relative importance of control, macro-scale, and micro-scale variables.

| OLS Diagnosis | Control Variables | Macro-Scale 5Ds | | | Micro-Scale |
|---|---|---|---|---|---|
| Buffer zone | - | 20 m | 50 m | 100 m | - |
| R2 | 0.014 | 0.366 | 0.372 | 0.374 | 0.146 |
| Adjusted R2 | 0.013 | 0.365 | 0.372 | 0.373 | 0.146 |
| F-statistic (sig.) | 110.341 *** | 748.906 *** | 770.332 *** | 774.241 *** | 255.843 *** |

Notes: *p*-value *** $p < 0.01$.

In the end, the explanatory variables that had the highest relative importance were macro-scale built environmental variables. The adjusted R2 values of the three buffers were 0.365, 0.372, and 0.373. The micro-scale variables ranked second in importance. The control variables' contribution was the smallest.

Since the OLS models with a 100 m buffer showed the highest goodness of fit, we only show the details of the OLS models with 100 m buffer in the tables. The results of OLS models 1–4 are shown in Tables 3 and 4.

The OLS models of the three buffers all showed severe collinearity for three variables, namely residential streets, annual mean NO2, and pixel ratios of buildings. After excluding these three variables, the VIFs of all other variables in the final OLS models were less than 10, which proved that there was no multicollinearity problem in our models. Moreover, the F statistics for all models were significant.

Comparing OLS models 1–4, we found that both sets of variables contribute uniquely to the explanatory power of the baseline model (Model 1) when either macro-scale built environment variables or micro-scale streetscape variables were added separately. However, the streetscape variables did not seem to provide substantial improvements in terms of $R^2$ when all three sets of variables were included in the analysis.

To further explore the reasons, we conducted a pairwise correlation matrix analysis between macro-scale 5Ds variables and micro-scale streetscape variables to verify the correlation between variables. We screened out pairwise variables with correlation coefficients greater than 0.2 for visualization. As can be seen from Figure 7, there were moderate or even high correlations between some variables. For example, in the SVIs variable, SVF had a moderate correlation with building density, PTALS, and per capita income (coefficient > 0.3), and the correlation between car and street noise was also greater than 0.3. It is worth noting that the correlation between grass and open space area was up to 0.54. In addition, there was a relatively strong correlation between tree, canopy density, and open space area (0.44 and 0.48, respectively). This showed that in the micro-scale streetscape variable, there were indeed some street view features that overlapped with the 5Ds macro variable. This may be the reason why, when macro-scale variables or street view variables were added separately, they had higher explanatory power, but when the three groups of variables were combined, the R2 of the model did not improve significantly. In addition, due to the relatively strong correlation between some variables, to prevent the collinearity effect on the results of the OLS models, we further examined how the results changed

when using different variable selection methods (e.g., stepwise regression, eliminating unimportant variables, or lowering the VIF threshold again to 3). However, regardless of the variable selection method, the relative importance of macro-scale features and street view factors in model fitting did not change.

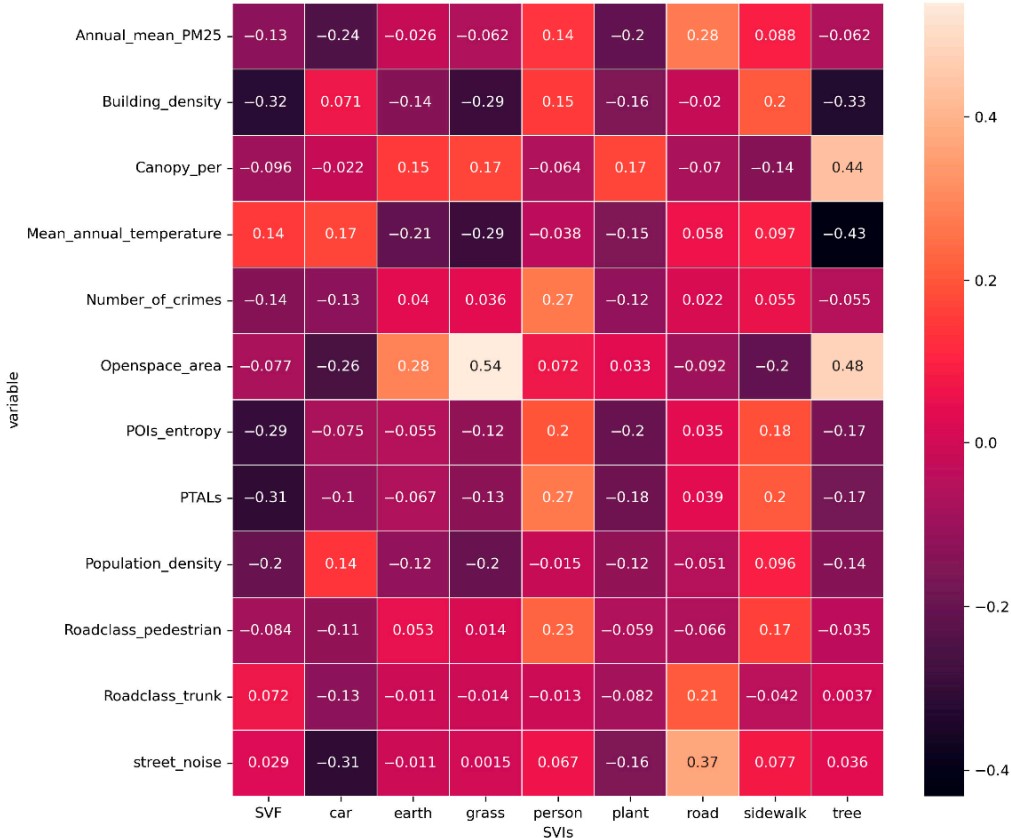

**Figure 7.** The pairwise correlation matrix analysis between macro-scale variables and micro-scale variables (only pairs of variables with correlation coefficients greater than 0.2 were visualized).

**Table 3.** The ordinary least squares (OLS) in a 100 m buffer zone of 40,290 sample points.

| Variables | OLS Model 1 | | OLS Model 2 | | OLS Model 3 | |
|---|---|---|---|---|---|---|
| | Coef. (Std. Dev.) | β | Coef. (Std. Dev.) | β | Coef. (Std. Dev.) | β |
| Control variables | | | | | | |
| Age groups | | | | | | |
| Pop0to17 | −12.521 *** (0.573) | −0.192 | −14.097 *** (0.532) | −0.216 | −10.833 *** (0.548) | −0.166 |
| Pop18to44 | 2.186 *** (0.410) | 0.034 | 3.475 *** (0.371) | 0.053 | 1.395 *** (0.392) | 0.021 |
| Pop45to59 | 7.162 *** (0.607) | 0.110 | 5.798 *** (0.524) | 0.089 | 5.512 *** (0.569) | 0.085 |
| Pop over75 | 0.520 (0.376) | 0.008 | 0.317 (0.321) | 0.005 | −1.080 *** (0.353) | −0.017 |
| Per capita income 2019 | −0.251 (0.397) | −0.004 | −6.269 *** (0.626) | −0.096 | −0.667 * (0.382) | −0.010 |
| Macro-scale built environments 5Ds | | | | | | |
| Density | | | | | | |
| Population density 2020 | | | −3.016 *** (0.306) | −0.046 | | |
| Job density 2019 | | | 4.948 *** (0.505) | 0.076 | | |
| Building density | | | −5.329 *** (0.371) | −0.082 | | |

**Table 3.** *Cont.*

| Variables | OLS Model 1 | | OLS Model 2 | | OLS Model 3 | |
|---|---|---|---|---|---|---|
| | Coef. (Std. Dev.) | β | Coef. (Std. Dev.) | β | Coef. (Std. Dev.) | β |
| Diversity | | | | | | |
| Street type | | | | | | |
| Trunk road | | | 2.842 *** (0.397) | 0.044 | | |
| Primary road | | | 12.004 *** (0.348) | 0.184 | | |
| Secondary road | | | 10.910 *** (0.271) | 0.167 | | |
| Tertiary road | | | 13.562 *** (0.272) | 0.208 | | |
| Pedestrian street | | | −0.575 * (0.323) | −0.009 | | |
| Cycleway | | | 6.473 *** (0.348) | 0.099 | | |
| Footway | | | −1.207 ** (0.549) | −0.019 | | |
| Service street | | | −13.236 *** (0.379) | −0.203 | | |
| Track | | | −3.013 *** (0.261) | −0.046 | | |
| Path | | | −2.452 *** (0.289) | −0.038 | | |
| POI entropy | | | 3.068 *** (0.350) | 0.047 | | |
| Design | | | | | | |
| Design: street amenities | | | | | | |
| Open space area | | | 12.707 *** (0.341) | 0.195 | | |
| Canopy density | | | −2.580 *** (0.308) | −0.040 | | |
| Number of intersections | | | 2.878 *** (0.356) | 0.044 | | |
| Number of traffic lights | | | 0.675 * (0.371) | 0.010 | | |
| Number of parking lots | | | −1.401 *** (0.261) | −0.021 | | |
| Maximum speed | | | −12.965 *** (0.637) | −0.199 | | |
| Street length | | | 8.372 *** (0.274) | 0.128 | | |
| Design: safety | | | | | | |
| Number of crimes | | | −3.966 *** (0.357) | −0.061 | | |
| Number of traffic accidents | | | 2.094 *** (0.320) | 0.032 | | |
| Number of fires | | | 0.016 (0.354) | 0.000 | | |
| Design: level of street pollution | | | | | | |
| Annual mean PM2.5 | | | −0.905 ** (0.404) | −0.014 | | |
| Street noise pollution level | | | 6.765 *** (0.420) | 0.104 | | |
| Design: street environment attributes | | | | | | |
| Street slope | | | 0.589 ** (0.265) | 0.009 | | |
| Annual mean temperature | | | −4.698 *** (0.327) | −0.072 | | |
| Destination accessibility | | | | | | |
| BtA800 | | | 3.708 *** (0.296) | 0.057 | | |
| BtA6300 | | | 5.059 *** (0.310) | 0.078 | | |
| Distance to transit | | | | | | |
| PTALs | | | −2.658 *** (0.393) | −0.041 | | |
| Micro-scale built environments | | | | | | |
| Pixel ratios of wall | | | | | 2.819 *** (0.340) | 0.043 |
| Pixel ratios of tree | | | | | 15.234 *** (0.394) | 0.234 |
| Pixel ratios of road | | | | | 7.727 *** (0.397) | 0.119 |
| Pixel ratios of grass | | | | | 7.832 *** (0.424) | 0.120 |
| Pixel ratios of sidewalk | | | | | 8.765 *** (0.401) | 0.135 |
| Pixel ratios of earth | | | | | 3.914 *** (0.334) | 0.060 |
| Pixel ratios of plant | | | | | −1.872 *** (0.334) | −0.029 |
| Pixel ratios of car | | | | | −2.253 *** (0.432) | −0.035 |
| Pixel ratios of fence | | | | | −4.431 *** (0.326) | −0.068 |
| Pixel ratios of signboard | | | | | 1.839 *** (0.310) | 0.028 |
| Pixel ratios of awning | | | | | 1.190 *** (0.305) | 0.018 |
| Pixel ratios of streetlight | | | | | 2.363 *** (0.310) | 0.036 |
| Pixel ratios of ashcan | | | | | −0.339 (0.313) | −0.005 |
| Pixel ratios of railing | | | | | 2.893 *** (0.310) | 0.044 |
| Pixel ratios of person | | | | | 4.597 *** (0.318) | 0.071 |
| Pixel ratios of minibike | | | | | −0.039 (0.301) | −0.001 |
| Pixel ratios of sculpture | | | | | −0.262 (0.305) | −0.004 |
| Pixel ratios of bridge | | | | | −0.552 * (0.305) | −0.008 |
| Pixel ratios of fountain | | | | | −0.011 (0.305) | 0.000 |
| Pixel ratios of windowpane | | | | | 0.253 (0.303) | 0.004 |

**Table 3.** *Cont.*

| Variables | OLS Model 1 | | OLS Model 2 | | OLS Model 3 | |
|---|---|---|---|---|---|---|
| | Coef. (Std. Dev.) | β | Coef. (Std. Dev.) | β | Coef. (Std. Dev.) | β |
| Pixel ratios of mountain | | | | | 0.377 (0.299) | 0.006 |
| Pixel ratios of water | | | | | 8.131 *** (0.317) | 0.125 |
| Pixel ratios of booth | | | | | 1.031 *** (0.300) | 0.016 |
| Pixel ratios of skyscraper | | | | | 1.297 *** (0.302) | 0.020 |
| Pixel ratios of lake | | | | | 0.231 (0.299) | 0.004 |
| Pixel ratios of pier | | | | | 1.882 *** (0.307) | 0.029 |
| SVF | | | | | 7.600 *** (0.390) | 0.117 |
| (Constant) | 78.736 *** (0.322) | | 78.736 *** (0.255) | | 78.736 *** (0.298) | |
| $R^2$ | 0.014 | | 0.386 | | 0.155 | |
| Adjusted $R^2$ | 0.013 | | 0.385 | | 0.155 | |
| F-statistic (sig.) | 110.341 *** | | 701.841 *** | | 231.589 *** | |

Coef. = unstandardized coefficient; Std. Dev.= standard error. β = standardized coefficients. Significance levels: * $p < 0.1$. ** $p < 0.05$. *** $p < 0.01$.

*4.4. Moran's I Test and Spatial Model Results*

A local indicators of spatial association (LISA) cluster map can map the clustering phenomenon of running hot spots. As can be seen from the LISA cluster map of the running amount (Figure 8), running hot spots in Inner London mainly appeared along the Thames River and in large parks or open areas, where high–high values gather. Running cold spots (low–low values) mainly appeared in South Kensington (a) A, located in the south of Hyde Park; Paddington (a) B, in the north of Hyde Park; Stepney (a) C, in the south of the Victoria Park; Forest Gate (a) D, located in the east of the Olympic Park; Romford (a) E, located in Inner London's northeast; and other regions. These areas are often densely populated with houses and dwellings.

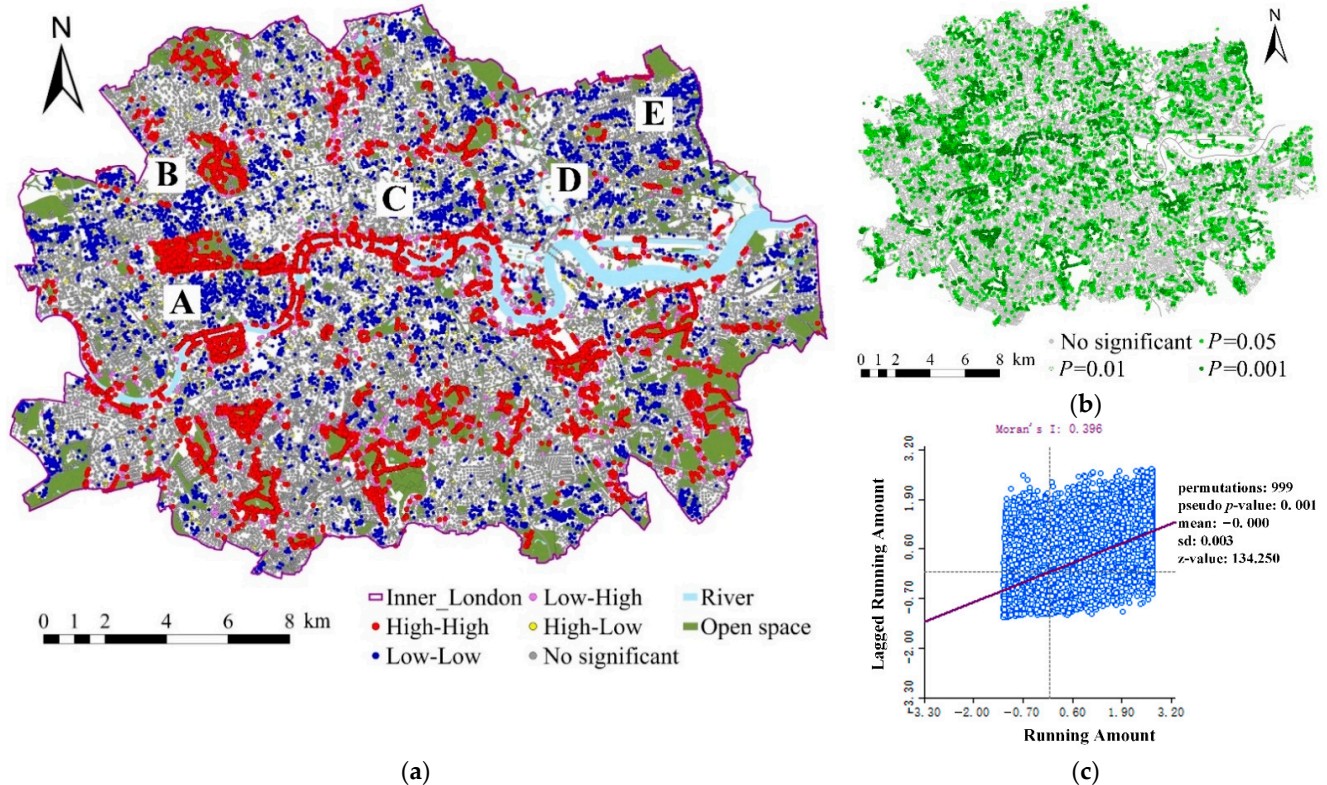

**Figure 8.** (**a**) LISA cluster map of the running amount based on 40,290 sample points, position A–E indicates the representative area where low–low values appear; (**b**) the LISA significance level map; (**c**) Moran's I test of running amount, which showed a positive spatial autocorrelation.

We performed a Moran's I test (with 999 permutations) on residual errors in Model 4, and the spatial weight matrix W used the "rook" method. The result showed that the Moran's I value on OLS residuals was 0.354, *p*-value = 0.001 < 0.05 (Figure 9a). This indicates a significant positive spatial autocorrelation on the OLS residuals. Moreover, the robust Lagrange multiplier (lag) and robust Lagrange multiplier (error) were both significant (Table 4), indicating the existence of both spatial lag and error effects. The SAC model can combine the spatially lagged dependent variable with the spatially lagged error term in the spatial modeling process to account for spatial interactions, so it was chosen. Moran's I value on the SAC model residual was −0.001, *p* = 0.333 >0.05 (Figure 9b), which means that the spatial autocorrelation was not significant, and the SAC model dealt with spatial autocorrelation well. In addition, compared with the OLS model, the Moran's I value of the SAC model's residuals decreased significantly. The R2 value was also greatly improved from 0.411 (Model 4) to 0.619 (SAC Model), so we would discuss the results based on the SAC results.

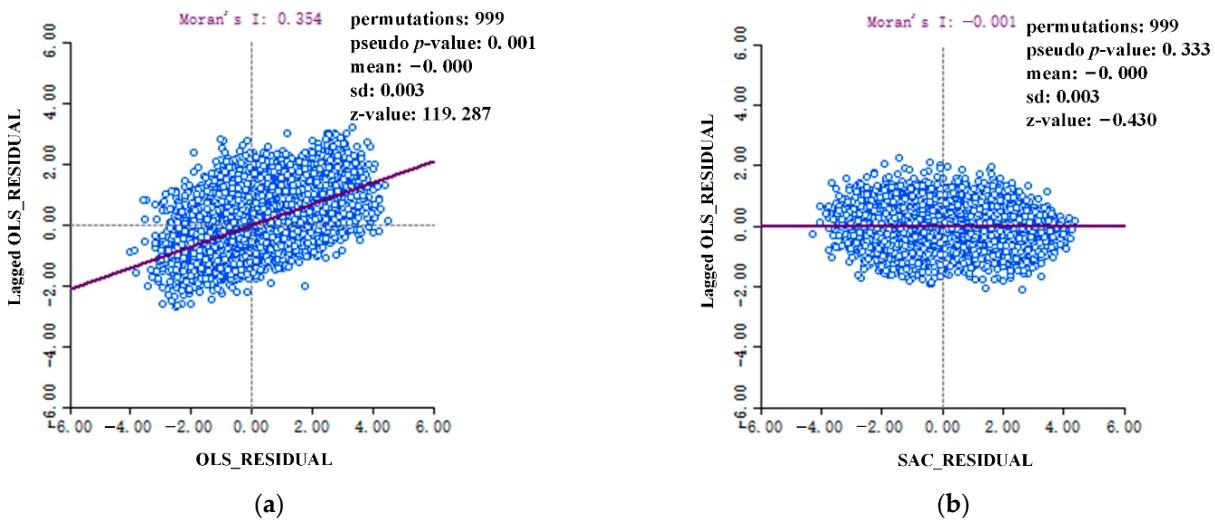

**Figure 9.** The results of Moran's I test. (**a**) The spatial autocorrelation of residual error of the OLS model (100 m buffer); (**b**) the counterpart of residual error of the SAC model (100 m buffer).

**Table 4.** The ordinary least squares results (OLS) and spatial regression results (SAC).

| Variables | OLS Model4 | | | | SAC Model | |
|---|---|---|---|---|---|---|
| | Coef. (Std. Dev.) | β | Sig. | VIF | Coef. (Std. Dev.) | Sig. |
| Dependent variable<br>  Running amount | | | | | | |
| Control variables | | | | | | |
| Age groups | | | | | | |
|   Pop0to17 | −13.632 *** (0.526) | −0.209 | 0.000 | 4.444 | 0.089 (0.265) | 0.737 |
|   Pop18to44 | 3.379 *** (0.367) | 0.052 | 0.000 | 2.169 | −0.247 (0.181) | 0.174 |
|   Pop45to59 | 6.098 *** (0.517) | 0.094 | 0.000 | 4.303 | −0.232 (0.254) | 0.361 |
|   Pop over75 | 0.083 (0.316) | 0.001 | 0.792 | 1.608 | 0.418 *** (0.154) | 0.007 |
|   Per capita income2019 | −7.292 *** (0.622) | −0.112 | 0.000 | 6.230 | −0.743 ** (0.308) | 0.016 |
| Macro-scale built environments 5Ds | | | | | | |
| Density | | | | | | |
|   Population density2020 | −3.418 *** (0.306) | −0.052 | 0.000 | 1.502 | 0.211 (0.157) | 0.179 |
|   Job density2019 | 5.650 *** (0.503) | 0.087 | 0.000 | 4.070 | −0.968 *** (0.252) | 0.000 |
|   Building density | −4.645 *** (0.381) | −0.071 | 0.000 | 2.336 | −0.176 (0.202) | 0.383 |

**Table 4.** *Cont.*

| Variables | OLS Model4 | | | | SAC Model | |
|---|---|---|---|---|---|---|
| | Coef. (Std. Dev.) | β | Sig. | VIF | Coef. (Std. Dev.) | Sig. |
| Diversity | | | | | | |
| Street type | | | | | | |
| Trunk road | 2.885 *** (0.391) | 0.044 | 0.000 | 2.456 | 3.426 ***(0.273) | 0.000 |
| Primary road | 11.479 *** (0.342) | 0.176 | 0.000 | 1.884 | 8.785 ***(0.245) | 0.000 |
| Secondary road | 10.310 *** (0.268) | 0.158 | 0.000 | 1.158 | 6.683 ***(0.186) | 0.000 |
| Tertiary road | 12.975 *** (0.269) | 0.199 | 0.000 | 1.165 | 8.439 ***(0.185) | 0.000 |
| Pedestrian street | −2.029 *** (0.323) | −0.031 | 0.000 | 1.682 | −0.863 ***(0.215) | 0.000 |
| Cycleway | 6.006 *** (0.344) | 0.092 | 0.000 | 1.899 | 5.155 ***(0.232) | 0.000 |
| Footway | −1.579 *** (0.541) | −0.024 | 0.004 | 4.712 | 0.900 **(0.354) | 0.011 |
| Service street | −12.418 *** (0.376) | −0.191 | 0.000 | 2.274 | −6.866 *** (0.254) | 0.000 |
| Track | −2.875 *** (0.256) | −0.044 | 0.000 | 1.054 | −1.162 *** (0.177) | 0.000 |
| Path | −2.422 *** (0.285) | −0.037 | 0.000 | 1.303 | −1.190 *** (0.193) | 0.000 |
| POI entropy | 2.652 *** (0.348) | 0.041 | 0.000 | 1.944 | −0.536 *** (0.197) | 0.007 |
| Design | | | | | | |
| Design: street amenities | | | | | | |
| Open space area | 12.391 *** (0.379) | 0.190 | 0.000 | 2.309 | 1.698 *** (0.212) | 0.000 |
| Canopy density | −2.868 *** (0.313) | −0.044 | 0.000 | 1.575 | −1.061 *** (0.168) | 0.000 |
| Number of intersections | 2.712 *** (0.351) | 0.042 | 0.000 | 1.984 | 0.689 *** (0.198) | 0.001 |
| Number of traffic lights | 0.764 ** (0.364) | 0.012 | 0.036 | 2.137 | −0.417 ** (0.207) | 0.044 |
| Number of parking lots | −1.249 *** (0.256) | −0.019 | 0.000 | 1.053 | −0.312 ** (0.146) | 0.032 |
| Maximum speed | −12.174 *** (0.627) | −0.187 | 0.000 | 6.323 | −4.320 *** (0.405) | 0.000 |
| Street length | 8.198 *** (0.271) | 0.126 | 0.000 | 1.180 | 4.395 *** (0.176) | 0.000 |
| Design: safety | | | | | | |
| Number of crimes | −4.349 *** (0.356) | −0.067 | 0.000 | 2.037 | −0.689 *** (0.180) | 0.000 |
| Number of traffic accidents | 1.809 *** (0.315) | 0.028 | 0.000 | 1.594 | 0.370 ** (0.181) | 0.040 |
| Number of fires | −0.326 (0.350) | −0.005 | 0.350 | 1.966 | −1.018 *** (0.170) | 0.000 |
| Design: level of street pollution | | | | | | |
| Annual mean PM2.5 | −0.555 (0.400) | −0.009 | 0.166 | 2.578 | −1.239 *** (0.259) | 0.000 |
| Street noise pollution level | 6.027 *** (0.425) | 0.092 | 0.000 | 2.908 | 1.239 *** (0.267) | 0.000 |
| Design: street environment attributes | | | | | | |
| Street slope | 0.568 ** (0.260) | 0.009 | 0.029 | 1.090 | 0.021 (0.169) | 0.901 |
| Annual mean temperature | −4.106 *** (0.327) | −0.063 | 0.000 | 1.724 | 0.553 *** (0.194) | 0.004 |
| Destination accessibility | | | | | | |
| BtA800 | 3.176 *** (0.292) | 0.049 | 0.000 | 1.368 | 2.866 *** (0.195) | 0.000 |
| BtA6300 | 4.960 *** (0.303) | 0.076 | 0.000 | 1.481 | 2.861 *** (0.214) | 0.000 |
| Distance to transit | | | | | | |
| PTALs | −3.158 *** (0.390) | −0.048 | 0.000 | 2.444 | −0.646 *** (0.199) | 0.001 |
| Micro-scale built environments | | | | | | |
| Pixel ratios of wall | −0.182 (0.293) | −0.003 | 0.535 | 1.377 | −1.119 *** (0.187) | 0.000 |
| Pixel ratios of tree | 4.729 *** (0.381) | 0.073 | 0.000 | 2.340 | 1.724 *** (0.242) | 0.000 |
| Pixel ratios of road | 1.084 *** (0.348) | 0.017 | 0.002 | 1.953 | 1.300 *** (0.231) | 0.000 |
| Pixel ratios of grass | 0.290 (0.382) | 0.004 | 0.448 | 2.352 | −0.215 (0.240) | 0.369 |
| Pixel ratios of sidewalk | 6.183 *** (0.343) | 0.095 | 0.000 | 1.894 | 3.172 *** (0.229) | 0.000 |
| Pixel ratios of earth | 0.646 ** (0.287) | 0.010 | 0.024 | 1.325 | 0.205 (0.180) | 0.254 |
| Pixel ratios of plant | −1.740 *** (0.287) | −0.027 | 0.000 | 1.324 | −0.664 *** (0.192) | 0.001 |
| Pixel ratios of car | 2.374 *** (0.372) | 0.036 | 0.000 | 2.221 | 0.507 ** (0.255) | 0.047 |
| Pixel ratios of fence | −3.062 *** (0.276) | −0.047 | 0.000 | 1.224 | −1.264 *** (0.191) | 0.000 |
| Pixel ratios of signboard | 0.428 (0.263) | 0.007 | 0.104 | 1.111 | 0.770 *** (0.193) | 0.000 |
| Pixel ratios of awning | 0.337 (0.256) | 0.005 | 0.187 | 1.054 | 0.409 ** (0.184) | 0.026 |
| Pixel ratios of streetlight | 1.699 *** (0.263) | 0.026 | 0.000 | 1.112 | 0.505 *** (0.185) | 0.006 |
| Pixel ratios of ashcan | −0.854 *** (0.265) | −0.013 | 0.001 | 1.133 | 0.117 (0.178) | 0.512 |
| Pixel ratios of railing | 0.798 *** (0.263) | 0.012 | 0.002 | 1.111 | 0.101 (0.177) | 0.570 |
| Pixel ratios of person | 2.827 *** (0.278) | 0.043 | 0.000 | 1.245 | 0.945 *** (0.182) | 0.000 |
| Pixel ratios of minibike | −0.250 (0.252) | −0.004 | 0.321 | 1.024 | −0.098 (0.191) | 0.609 |
| Pixel ratios of sculpture | −0.005 (0.255) | 0.000 | 0.986 | 1.049 | −0.065 (0.177) | 0.712 |
| Pixel ratios of bridge | −1.006 *** (0.257) | −0.015 | 0.000 | 1.061 | −0.574 *** (0.174) | 0.001 |
| Pixel ratios of fountain | −0.207 (0.255) | −0.003 | 0.417 | 1.045 | −0.180 (0.186) | 0.335 |
| Pixel ratios of windowpane | −0.081 (0.254) | −0.001 | 0.750 | 1.034 | −0.127 (0.192) | 0.509 |
| Pixel ratios of mountain | 0.117 (0.250) | 0.002 | 0.640 | 1.006 | 0.432 ** (0.188) | 0.022 |
| Pixel ratios of water | 6.153 *** (0.267) | 0.094 | 0.000 | 1.144 | 2.423 *** (0.178) | 0.000 |
| Pixel ratios of booth | 0.364 (0.251) | 0.006 | 0.147 | 1.013 | 0.599 *** (0.193) | 0.002 |
| Pixel ratios of skyscraper | 0.715 *** (0.253) | 0.011 | 0.005 | 1.028 | −0.132 (0.162) | 0.417 |

**Table 4.** *Cont.*

| Variables | OLS Model4 | | | | SAC Model | |
|---|---|---|---|---|---|---|
| | Coef. (Std. Dev.) | β | Sig. | VIF | Coef. (Std. Dev.) | Sig. |
| Pixel ratios of lake | −0.055 (0.250) | −0.001 | 0.827 | 1.009 | 0.135 (0.168) | 0.420 |
| Pixel ratios of pier | 1.670 *** (0.257) | 0.026 | 0.000 | 1.063 | 0.214 (0.184) | 0.243 |
| SVF | 2.021 *** (0.363) | 0.031 | 0.000 | 2.115 | 0.584 ** (0.232) | 0.012 |
| (Constant) | 78.736 *** (0.249) | | 0.000 | | 3.705 *** (0.317) | 0.010 |
| Wy | | | | | 0.956 *** (0.004) | 0.000 |
| LAMBDA | | | | | −0.707 *** (0.009) | 0.000 |
| $R^2$ | 0.411 | | | | 0.619 | |
| Adjusted $R^2$ | 0.410 | | | | | |
| F-statistic (sig.) | 445.783 *** | | | | | |
| Moran's I on residuals (z-value) | 0.354 *** (119.287) | | | | −0.001(−0.430) | |
| Robust Lagrange multiplier (lag) | 111.835 *** | | | | | |
| Robust Lagrange multiplier (error) | 2694.828 *** | | | | | |

Coef. = unstandardized coefficient; Std. Dev.= standard error. β = standardized coefficients. Significance levels: **$p < 0.05$. ***$p < 0.01$.

## 5. Discussion

In this study, we used crowdsourced GPS track run data published on the Strava platform and other open-source datasets to reveal the spatial distribution and influencing factors of running activities in Inner London. We used emerging GSV images and DL technology to measure the micro-scale street-level built environment features as a supplement to the traditional 5Ds macro-scale variables. This paper discussed the specific influence of built environment features at different scales on runners' route preferences and the internal relationship between macro-scale built environment factors and the micro street view environment features.

### 5.1. Research Findings

The present study mainly answers four questions as follows:

*Question 1: How and which macro-scale built environment attributes that are classified based on the 5Ds framework influence the running activity in Inner London?*

According to the SAC model results, we found that some macro-scale built environment features have a significant effect on the running amount.

(1) For the density dimension, population density and building density did not show a correlation with the running amount, while job density indicated a negative relationship with running. In this respect, Ettema [11] found that high density did not seem to affect running participation. Places with a higher population and building density were often more urbanized and closer to downtown [10]. Although high density stimulates walking, it likely reduces the attractiveness of running because it causes many interactions with other road users and does not allow runners to maintain their momentum. In addition, the areas with higher job density tend to be central business districts (CBD), office spaces, and downtown areas in Inner London. These areas tend to be highly commercial and artificial, which may not be ideal for runners [10].

(2) For the diversity dimension, running activity occurred more frequently on trunk, primary, secondary, and tertiary roads, cycleways, and footways. This might be because trunk, primary, secondary, and tertiary roads tend to have strong connectivity, complete infrastructure, and wider street space. Cycleways and footways tend to have a more comfortable atmosphere for physical activity. On the contrary, runners are less likely to choose tracks (often rough with unpaved surfaces, for mostly agricultural or forestry uses), paths (for non-specific or shared use), pedestrian streets (used mainly for pedestrians in shopping and some residential areas), and service streets (for access roads to, or within, an industrial estate, campsite, business park, car park, alleys, etc.) for their running activities compared to other road types (Figure 10). The possible reasons might be that unpaved surfaces, less wide paths, and busy streets with many

pedestrians or heavy traffic volume are proven to be the features that have negative impacts on running satisfaction or frequency [10,11,13]. This is also in line with the results of [71], which reported that a comfortable running surface was important for runners and had a positive effect.

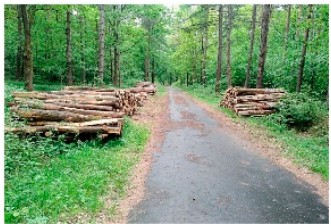 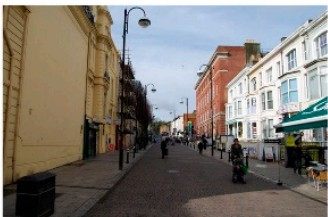 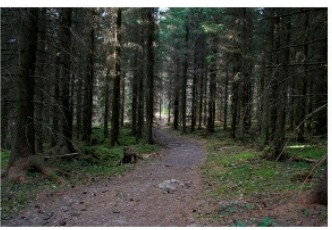 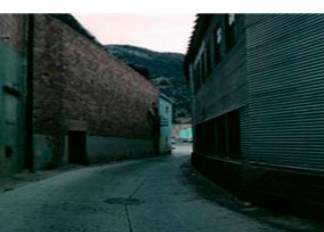

| Track | Pedestrian street | Path | Service street |

**Figure 10.** The four road types are negatively correlated with running. These images corresponding to these road types are from the OSM road classification document.

The POI entropy also showed a negative correlation with running. Areas with a higher POI entropy are associated with a higher degree of functional mixing; in other words, those areas tend to be more urbanized. Previous studies have indicated that runners are more likely to run in environments that are closer to nature and away from the downtown core [6,10,11,21]. For example, Bodin and Hartig [21] suggested that runners indeed prefer green running environments over urban settings and report that they are better at offering fascination and escape from daily hassles. Additionally, the number of POIs was positively related to the running amount in Boston [32], which might be due to the difference in differences in scale and density between the two cities [60].

(3)  For the design dimension, the factors that showed positive impacts on running and promoted running hot spots were close to urban open spaces, relatively long street segments, and higher safety. Many studies have provided similar results [6,10,11,27]. For instance, Shipway and Holloway [27] showed that runners preferred green, open, and natural running environments.

The design attributes that hindered running in our study were the canopy density, number of traffic lights and parking lots, maximum speed, number of crimes and fires, and annual mean PM2.5. Streets with a higher canopy density might give people a feeling of enclosure and depression, and thus might negatively affect running. Moreover, streets with high car speeds and areas with a large number of parking lots and traffic lights tend to have high volumes of traffic and give a perception of insecurity; thus, impeding running activity. An area with higher fire or crime frequency could also discourage running because of the perceived insecurity associated with these streets. Safety has been widely recognized to have significant impacts on runnability [5,10]. Moreover, street environments with higher levels of PM 2.5 can hinder running. According to previous studies, air pollution might affect the respiratory system during running and lead to an uncomfortable running experience, which can cause health problems [12,14,60].

However, our results showed that running hotspots occurred on streets with a high number of traffic accidents and in areas with higher street noise levels. This defies common sense, possibly because running is often attached to the road, so runners are often passively exposed to noise and traffic insecurity. This suggests that urban designers need to consider making running environments safer for runners in the future.

(4)  For the destination accessibility dimension, in sDNA analysis with a radius of 800 m (Figure 11a), the downtown area in the center of Inner London and some street networks in the northeast region have good accessibility. In sDNA analysis with a radius of 6300 m, important highways and main streets were identified to be highly accessible (Figure 11b). The SAC results showed that both BTA800 and BTA6300 were significantly positively correlated with running, suggesting that streets with

higher accessibility were more likely to attract runners. To our knowledge, this is the first study to use spatial syntax to measure the association between running activity and street accessibility. Previous studies have reported that streets with high accessibility promote walking [33]. The results of this study indicate that streets with high accessibility also promote running.

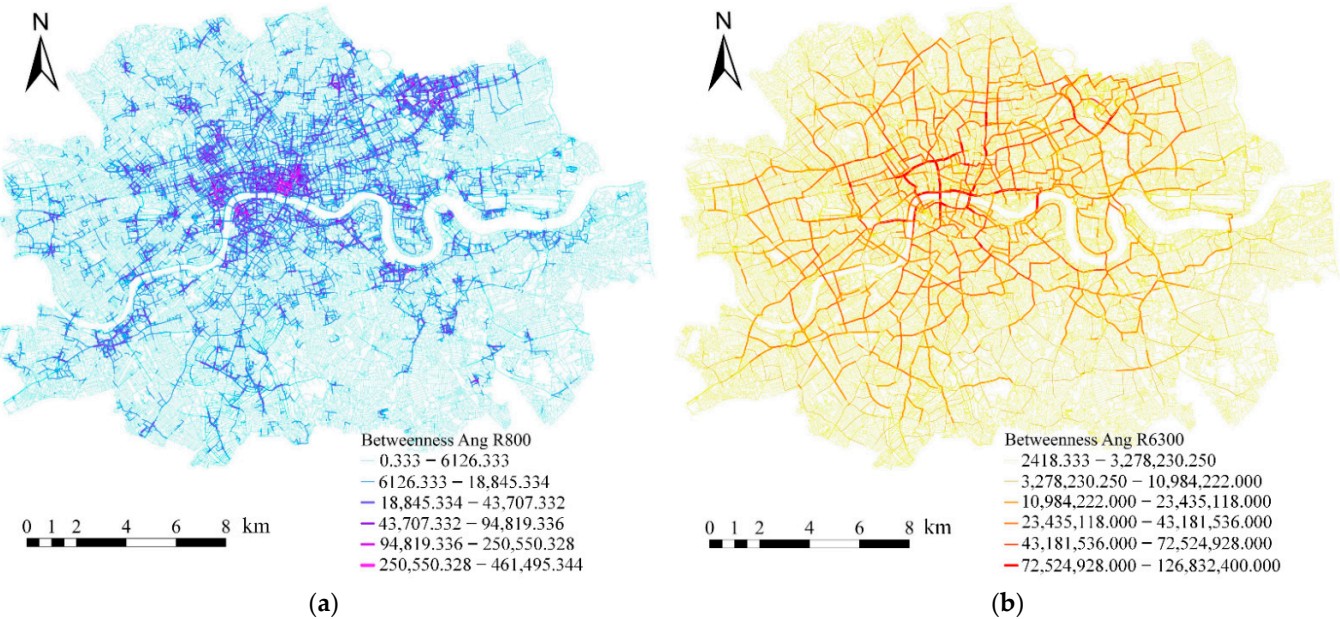

**Figure 11.** The sDNA space syntax analysis. (**a**) with an 800 m radius; (**b**) with a 6300 m radius.

(5)  For the distance to transit dimension, PTALs showed a negative correlation with running frequency; that is, runners were less likely to pass public transport service stations in Inner London. This is contrary to previous studies, which indicated that more public transport nodes and transportation facilities could promote running [6,32]. Inner London is a densely populated metropolis with dense and developed traffic. Better access to public transportation could indicate intense urbanization. Therefore, it is logical to assume that runners are more willing to stay away from heavy traffic areas. In addition, Ettema [11] regarded running as an activity similar to recreational (leisure) walking, and in this sense, minimizing the distance to the transportation facilities is not the goal of recreational walking or running. Running may be more of a pure form of exercise and physical activity, which is different from walking or cycling that often entail transferring traffic to other destinations.

*Question 2: How and which micro-scale streetscape features influence running amount?*

We found that some micro-scale street features play a crucial role in influencing human running behavior. Our results showed that hot spots for running activities were more likely to be in areas with wider roads and, especially, wider sidewalks, more trees, higher sky openness, more streetlights, and proximity to natural landscapes (e.g., mountains and water). In addition, some street services (e.g., booths, awnings, and signboards) also showed a positive effect on running. However, GSV features that hinder running were more architectural interfaces and fences, which reflect the enclosed degree of the streets. Previous studies have shown that people prefer to be physically active in environments that are more open, less artificial, and more natural [5].

Notably, pixel ratios of plants showed a negative correlation with running, while trees showed a positive correlation. In the PSPNet classification labels, plant refers to a shrub with a low height and branch points. A large number of shrubs may obstruct a runner's passage or obscure their view. Moreover, a dense brush can create dark shadow areas, and

studies have shown that dense brush areas are a potential risk for crime, which can make runners feel depressed and unsafe [72].

However, some of our results contradict conventional wisdom. The pixel ratios of car and person showed a positive correlation trend, which was different from previous studies. For instance, Ettema [11] showed that the presence of too many cars and pedestrians can become an obstacle for runners and disrupt the flow of running. This study showed the opposite. It may be that running is more dependent on the streets, so it is difficult to avoid the gathering of cars and pedestrians. In particular, among the 40,290 sample points, nearly 18,192 sample points were located on residential roads, which may have more pedestrians and stopped cars on the streets (Figure 12).

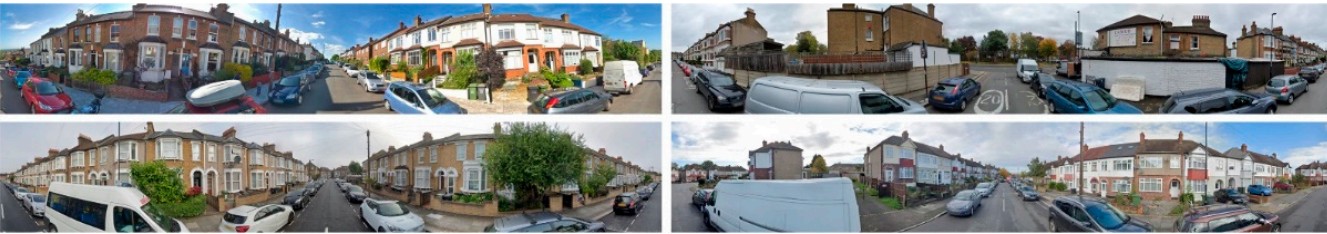

**Figure 12.** Residential streets are often occupied by many residents' cars in Inner London.

*Question 3: How do micro-scale streetscape features complement or conflict with macro-scale built environment indexes?*

By comparing the variable groups, the OLS results showed that both macro-scale and micro-scale variables showed statistical significance with the running amount. The variable group with higher relative importance was the macro-scale variables.

Furthermore, after comparing the OLS models 1–4, we found that micro-scale factors contributed strongly to the explanatory power of running amount, but when all three variable groups were included, GSV variables did not seem to provide substantial improvements to the R2. The pairwise correlation matrix analysis showed moderate or even high correlations between some variables. This showed that, in GSV variables, there were indeed some street view features that overlapped with the 5Ds macro variables. As a result, when the GSV variables and 5Ds macro variables entered the regression model at the same time, there would be some overlap in the explanatory power of the running amount. Our research supports that the GSV variables are a supplement to the 5Ds framework in built environment studies. First, when there are limitations to obtaining some macro-scale data, strongly correlated GSV variables can be considered as a substitute. Second, among GSV variables, some street view features do not correlate with macro 5Ds factors but with running behavior, and these features could be used as a supplement to the 5Ds framework to measure human running. However, the premise is that care should be taken to prevent the correlation between variables from causing multicollinearity effects on the models. No dataset can comprehensively measure all physical activities, but the combination and complement of these multi-source datasets may more comprehensively explain and measure the unknown laws; thus, providing useful guidance for urban construction.

*Question 4: Does the running amount in Inner London have spatial dependence effects?*

Spatial autocorrelation causes interference in the OLS models [18,61,64,66]. To obtain robust results, it is necessary to study the spatial effects of the model. A significant spatial autocorrelation effect was indeed detected in the residuals of our OLS model. We solved the problem of spatial dependence well by establishing the SAC model. In addition, compared with the regression results of the OLS model and SAC model, although the signs and significance of most variables remain unchanged after regression, some variables do show deviations. Therefore, taking the spatial effect into account can help us to better understand the spatial distribution regularities of running and make our regression results more reliable.

### 5.2. Practical Implications

Our findings enrich the literature on the built environment and human behaviors, and provide empirical evidence for urban designers to develop a running-friendly built environment, in turn encouraging residents to run, which can improve public health.

First of all, our study revealed that macro-scale built environment variables and micro-scale GSV variables both have a significant effect on the running amount. Therefore, city planners and landscape architects should not only consider the macro-scale built environment features but also eye-level street quality. For example, planners should seek to increase the width and length of roads (especially sidewalks), pay attention to the openness of running routes, add plants with higher branch points rather than dense bushes that block one's view, and reduce enclosures or excessive artificial atmospheres.

Second, we found that there was some overlap between the GSV variables and the 5Ds macro variables in explaining the running amount. Some GSV variables have a relatively high correlation with macro 5Ds variables. These highly correlated GSV features are potential substitutes for macro data (when access is limited), for example, open space and pixel ratios of grass and trees, level of street noise, and pixel ratios of cars, etc.

Third, we found that many built environment features (e.g., open space areas, street accessibility, street length, street safety, level of PM2.5, etc.) might promote or hinder running behavior. Adjustments to the built environment and improvements to running routes could benefit many people and lead to more people participating in and sticking with running. In this respect, our study could provide some practical suggestions for a high-density city from a runner's point of view. Urban designers should consider these practical factors for a better running environment.

### 5.3. Limitations

The following limitations need to be noted. First, running activities observed from Strava may not be able to fully represent all residents' running behaviors in Inner London. In the Strava platform, demographic details were hidden to protect user anonymity, so there is no way to know for sure how representative of all the runners in Inner London these massive GPS tracks are. Second, running information at an individual level was missing, such as the frequency and duration of running on an individual level. Future studies using different methods (e.g., questionnaires and interviews) could be conducted to obtain more detailed information of running behaviors. Third, the micro-scale built environment features from GSVs may not necessarily be fully representative of the real scene that runners see during exercise. For example, night vision could be captured to investigate running behavior at night. Lastly, we found that some macro-scale built environment features were strongly correlated with some GSV features. However, whether this relationship will differ in other cities needs to be explored in the future. GSV features might be identified as a good potential alternative to some of the macro-scale built environment features, providing more detailed and robust data.

### 6. Conclusions

This study contributes to a more comprehensive view of the relationship between running activity and the urban built environment of different scales. Based on the semi-open Strava data, we explained the spatial distribution and spatial clustering effects of running activities in Inner London. We elucidated the mechanism of the built environment at different scales as contributors to running activity. Our study complemented the effects of microscopic streetscape features on running activity using GSV images, CV, and DL. Furthermore, we examined the potential relationship between these micro-scale GSV features and the macro-scale built environment features. In addition, we analyzed the spatial autocorrelation effect of running using SAC models.

This study found that both the macro- and micro-scale built environment features have a significant influence on running. The micro-scale built environment features extracted from GSV images from a scale similar to human eyes can be a good supplement

to the traditional macro-scale built environment features. Many specific macro-scale built environment features (e.g., open space areas, street accessibility, street length, street safety, level of street pollution, etc.) and GSV features (e.g., SVF, roads, sidewalks, trees, wall, fence, streetlight, etc.) were found to promote or hinder running behavior.

With the expansion of population in the urban area, the results of our study could help to cope with the rising needs of urban residents for the city of proximity. We believe that the results will bring new inspiration to urban planning and public health studies, and can provide practical suggestions for the creation of running-friendly cities. As a result, our study might have potential benefits for sustainable, fair, high-quality, and healthy living by better understanding the way the built environment affects the mobility of people from the perspective of running.

**Author Contributions:** Conceptualization, Hongchao Jiang and Lin Dong; methodology, Hongchao Jiang and Lin Dong; software, Hongchao Jiang; validation, Hongchao Jiang and Lin Dong; formal analysis, Hongchao Jiang and Lin Dong; resources, Hongchao Jiang; data curation, Hongchao Jiang; writing—original draft preparation, Hongchao Jiang and Lin Dong; writing—review and editing, Hongchao Jiang and Lin Dong; visualization, Hongchao Jiang; supervision, Bing Qiu; project administration, Bing Qiu; funding acquisition, Bing Qiu All authors have read and agreed to the published version of the manuscript.

**Funding:** This research were funded by the National Natural Science Foundation of China (NSFC) General Project (Grant No. 31971721) and the Priority Academic Program Development of Jiangsu Higher Education Institutions (PAPD).

**Data Availability Statement:** The Inner London street network data presented in this study are openly available from [https://download.geofabrik.de/europe/great-britain/england/greater-london.html] (accessed on 20 February 2022). The Inner London boundary is from [https://data.london.gov.uk/dataset/inner-and-outer-london-boundaries-london-plan-consultation-2009] (accessed on 10 March 2022). The Strava Heatmap presented in this study is openly available from [https://www.strava.com/heatmap#11.40/-0.09927/51.52748/hot/run] (accessed on 4 April 2022). Population density (2020) and age group data are downloaded from [https://www.ons.gov.uk/peoplepopulationandcommunity/populationandmigration/populationestimates/datasets/censusoutputareaestimatesinthelondonregionofengland] (accessed on 12 March 2022). Job density data is from [https://data.london.gov.uk/dataset/jobs-and-job-density-borough] (accessed on 12 March 2022). Open space data is from [https://data.london.gov.uk/dataset/designated_open_space] (accessed on 12 March 2022). Tree canopy data comes from [https://data.london.gov.uk/dataset/curio-canopy] (accessed on 27 April 2022). Per capita income data is from [https://www.ons.gov.uk/economy/regionalaccounts/grossdisposablehouseholdincome/datasets/regionalgrossdisposablehouseholdincomelocalauthoritiesbyitl1region] (accessed on 12 March 2022). Crimes data is from [https://data.london.gov.uk/dataset/recorded_crime_summary] (accessed on 12 March 2022). Traffic accident data is from [https://data.gov.uk/dataset/cb7ae6f0-4be6-4935-9277-47e5ce24a11f/road-safety-data] (accessed on 25 April 2022). Fires facts dataset is from [https://data.london.gov.uk/dataset/lfb-fires-in-london-1966-2019---fire-facts] (accessed on 25 April 2022). London Atmospheric Emissions Inventory (LAEI) 2019 dataset is downloaded from [https://data.london.gov.uk/dataset/london-atmospheric-emissions-inventory--laei---2019] (accessed on 27 April 2022). The noise pollution in London dataset is downloaded from [https://data.london.gov.uk/dataset/noise-pollution-in-london] (accessed on 25 April 2022). The night-light data comes from [http://59.175.109.173:8888/index.html#] (accessed on 10 May 2022). DEM data is from the Rivermap software [http://www.rivermap.cn/home/mapdata.html] (accessed on 12 May 2022). Temperature data is from [https://data.london.gov.uk/dataset/major-summer-heatspots-using-landsat-8-thermal-satellite-data] (accessed on 25 April 2022). The public transport accessibility levels (PTALs) data is from [https://data.london.gov.uk/dataset/public-transport-accessibility-levels] (accessed on 25 April 2022). Building density, road types, intersections, traffic lights, parking lots, and POIs data are from [https://download.geofabrik.de/europe/great-britain/england/greater-london.html] (accessed on 20 February 2022).

**Conflicts of Interest:** The authors declare no conflict of interest.

## Appendix A

**Table A1.** The descriptive statistics of all variables.

| Variables | 20 m Buffer | 50 m Buffer | 100 m Buffer | N |
|---|---|---|---|---|
| | Mean (S.D.) | Mean (S.D.) | Mean (S.D.) | |
| **Dependent variable** | | | | |
| Running amount | 78.736 (65.165) | | | 40,290 |
| **Control variables** | | | | |
| Age groups | | | | |
| Pop0to17 | 3258.119 (1097.110) | | | 40,290 |
| Pop18to44 | 7345.544 (3118.943) | | | 40,290 |
| Pop45to59 | 2685.136 (624.544) | | | 40,290 |
| Pop60to74 | 1451.455 (313.130) | | | 40,290 |
| Pop over75 | 652.967 (199.145) | | | 40,290 |
| Per capita income 2019 | 35,659.353 (21754.988) | | | 40,290 |
| **Macro-scale built environments 5Ds** | | | | |
| Density | | | | |
| Population density 2020 | 11,064.400 (5345.130) | | | 40,290 |
| Job density 2019 | 2.172 (10.041) | | | 40,290 |
| Building density | 0.152 (0.165) | 0.188 (0.156) | 0.193 (0.143) | 40,290 |
| Diversity | | | | |
| Street type | | | | |
| Trunk road | (0. no; 1. yes) | | | 1825 |
| Primary road | (0. no; 1. yes) | | | 2172 |
| Secondary road | (0. no; 1. yes) | | | 1141 |
| Tertiary road | (0. no; 1. yes) | | | 2546 |
| Residential street | (0. no; 1. yes) | | | 18,192 |
| Living street | (0. no; 1. yes) | | | 101 |
| Pedestrian street | (0. no; 1. yes) | | | 1032 |
| Cycleway | (0. no; 1. yes) | | | 1411 |
| Footway | (0. no; 1. yes) | | | 8287 |
| Service street | (0. no; 1. yes) | | | 3023 |
| Track | (0. no; 1. yes) | | | 85 |
| Path | (0. no; 1. yes) | | | 475 |
| POI entropy | 0.899 (0.897) | | | 40,290 |
| Design | | | | |
| Design: street amenities | | | | |
| Open space area | 147.831 (345.982) | 977.392 (2017.663) | 4148.766 (7474.492) | 40,290 |
| Canopy density | 16.906 (11.340) | | | |
| Number of intersections | 0.141 (0.492) | 0.538 (1.194) | 1.754 (2.618) | 40,290 |
| Number of traffic lights | 0.051 (0.313) | 0.225 (0.858) | 0.742 (1.831) | 40,290 |
| Number of parking lots | 0.001(0.033) | 0.008(0.099) | 0.033(0.219) | 40,290 |
| Maximum speed | 21.921 (17.978) | | | 40,290 |
| Street length | 0.321 (0.243) | | | 40,290 |
| Design: safety | | | | |
| Number of crimes | 3346.523 (3164.750) | | | 40,290 |
| Number of traffic accidents | 0.090 (0.387) | 0.339 (0.895) | 1.104 (1.919) | 40,290 |
| Number of fires | 632.301 (158.028) | | | 40,290 |
| Design: level of street pollution | | | | |
| Annual mean NO2 | 33.580 (9.184) | | | 40,290 |
| Annual mean PM2.5 | 11.658 (1.502) | | | 40,290 |
| Street noise pollution level | 3.117(1.768) | | | 40,290 |
| Design: street environment attributes | | | | |
| Street slope | 3.045 (2.377) | | | 40,290 |
| Night-light intensity | 0.001 (0.002) | | | 40,290 |
| Annual mean temperature | 32.675 (1.550) | | | 40,290 |
| Destination accessibility | | | | |
| BtA800 | 3582.961 (7723.815) | | | 40,290 |
| BtA6300 | 2,530,509.635 (7,744,852.284) | | | 40,290 |
| Distance to transit | | | | |
| PTALs | 20.550 (19.088) | | | 40,290 |

**Table A1.** *Cont.*

| Variables | 20 m Buffer | 50 m Buffer | 100 m Buffer | N |
|---|---|---|---|---|
| | Mean (S.D.) | Mean (S.D.) | Mean (S.D.) | |
| Micro-scale built environments | | | | |
| Pixel ratios of wall | 0.030 (0.054) | | | 40,290 |
| Pixel ratios of building | 0.257 (0.157) | | | 40,290 |
| Pixel ratios of tree | 0.163 (0.125) | | | 40,290 |
| Pixel ratios of road | 0.157 (0.071) | | | 40,290 |
| Pixel ratios of grass | 0.022 (0.056) | | | 40,290 |
| Pixel ratios of sidewalk | 0.078 (0.049) | | | 40,290 |
| Pixel ratios of earth | 0.005 (0.023) | | | 40,290 |
| Pixel ratios of plant | 0.029 (0.039) | | | 40,290 |
| Pixel ratios of car | 0.055 (0.049) | | | 40,290 |
| Pixel ratios of fence | 0.017 (0.028) | | | 40,290 |
| Pixel ratios of signboard | 0.003 (0.005) | | | 40,290 |
| Pixel ratios of awning | 0.000 (0.002) | | | 40,290 |
| Pixel ratios of streetlight | 0.001 (0.001) | | | 40,290 |
| Pixel ratios of van | 0.003 (0.009) | | | 40,290 |
| Pixel ratios of ashcan | 0.002 (0.004) | | | 40,290 |
| Pixel ratios of railing | 0.004 (0.012) | | | 40,290 |
| Pixel ratios of person | 0.002 (0.007) | | | 40,290 |
| Pixel ratios of minibike | 0.000 (0.002) | | | 40,290 |
| Pixel ratios of chair | 0.000 (0.001) | | | 40,290 |
| Pixel ratios of sculpture | 0.000 (0.000) | | | 40,290 |
| Pixel ratios of bicycle | 0.000 (0.002) | | | 40,290 |
| Pixel ratios of column | 0.000 (0.002) | | | 40,290 |
| Pixel ratios of bridge | 0.000 (0.005) | | | 40,290 |
| Pixel ratios of fountain | 0.000 (0.000) | | | 40,290 |
| Pixel ratios of windowpane | 0.000 (0.002) | | | 40,290 |
| Pixel ratios of mountain | 0.000 (0.001) | | | 40,290 |
| Pixel ratios of water | 0.001 (0.009) | | | 40,290 |
| Pixel ratios of ceiling | 0.001 (0.016) | | | 40,290 |
| Pixel ratios of booth | 0.000 (0.000) | | | 40,290 |
| Pixel ratios of sofa | 0.000 (0.000) | | | 40,290 |
| Pixel ratios of lamp | 0.000 (0.000) | | | 40,290 |
| Pixel ratios of skyscraper | 0.000 (0.001) | | | 40,290 |
| Pixel ratios of lake | 0.000 (0.000) | | | 40,290 |
| Pixel ratios of bulletin board | 0.000 (0.000) | | | 40,290 |
| Pixel ratios of desk | 0.000 (0.000) | | | 40,290 |
| Pixel ratios of pier | 0.000 (0.000) | | | 40,290 |
| SVF | 0.642 (0.222) | | | 40,290 |

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
