# Peer review of "How Are Macro-Scale and Micro-Scale Built Environments Associated with Running Activity? The Application of Strava Data and Deep Learning in Inner London"

_ijgi, doi:10.3390/ijgi11100504_

Round 1
Reviewer 1 Report
The Research topic is relevant, interesting and innovative, according to the Urban Public Health scientific scenario and the walkable outcomes related to Healthy Lifestyles'r promotion. According to criteria like "density, diversity, design" you can find used input in papers like: Rebecchi A., et Al. Walkable environments and healthy urban moves: Urban context features assessment framework experienced in Milan. Sustainability (MDPI, Switzerland), 2019, 11(10), 2778. DOI: 10.3390/su11102778..
Another challenging topic related to the research purpose is the city of proximity, that probably can be cited in the conclusions.
Author Response
Dear reviewer,
Thank you very much for taking the time to review this manuscript! Please see the attachment.
Best regards!

Reviewer 2 Report
First of all, let me say that this is an excellent paper. It is impeccably researched and reported.
But, there is so much detail and information in the paper that it could be two papers. I would start by simplifying some of the methods and results description.
Also, this statement: For the diversity dimension, running activity occurred more frequently on trunk, primary, secondary, and tertiary roads, cycleways, and footways" seems to indicate that running activity occurs on ALL roads and paths.
Overall, this paper is an important contribution to the emerging literature on running but needs simplification and clarification.
Author Response
Dear reviewer,
Thanks very much for taking the time to review this manuscript! Please see the attachment
Best regards!

Reviewer 3 Report
By collecting 40,29014 sample points from Strava in London, the article investigated how macro-scale built environment featuresand micro-street-level factors affect residents' running behavior. The paper used a novel research method and large sample size, which has implications for the construction of running- friendly cities. The specific modifications are as follows:1. The logic of the literature review part needs to be further strengthened. For example, the position of "2.1. the 5Ds framework and built environment" can be adjusted, as it is not tigtly connected with section 2.2.
2. The literature review part needs to further clarify, for insance why it is necessary to study the influence of urban built environment on residents' running behavior from different spatial scales, and whether previous studies have discussed this issue.
3. The paper adopted running raster value as the y variable, but whether population density should be used as a control variable instead of just a variable in 5d, because the number of runners in areas with high population density may be significantly higher and thus affect the regression results.
4. The sample representativeness of Strava needs further explanation, including how many people in London use the app to record running trajectories and whether the usage rate has fluctuated significantly in the past two years.
Author Response
Dear reviewer,
Thanks very much for taking the time to review this manuscript! Please see the attachment.
Best regards!

Reviewer 4 Report
1. Before raising the research questions in Section 1.2, the academic rationale for asking these four questions needs to be briefly stated.
2. The coefficients for variables such as population density, building density, and open space area in Table 3 are too large. Why does this happen? Dimensionless processing may be required.
3. Where there is a high density of work, there are many runners. Is this phenomenon because the data used by the author identifies people who walk to work as exercising or running?
Author Response

(The authors gave the same response as above.)

Round 2
Reviewer 3 Report
it addressed comments well and it could be accepted with present form.